Corrected: Publisher correction

# Multicolor multiscale brain imaging with chromatic multiphoton serial microscopy

Lamiae Abdeladim[1], Katherine S. Matho[1,2,3], Solène Clavreul [2], Pierre Mahou[1], Jean-Marc Sintes[1], Xavier Solinas[1], Ignacio Arganda-Carreras [4,5,6], Stephen G. Turney[7], Jeff W. Lichtman [7], Anatole Chessel [1], Alexis-Pierre Bemelmans [8], Karine Loulier [2], Willy Supatto[1], Jean Livet [2] & Emmanuel Beaurepaire [1]

Large-scale microscopy approaches are transforming brain imaging, but currently lack efficient multicolor contrast modalities. We introduce chromatic multiphoton serial (ChroMS) microscopy, a method integrating one-shot multicolor multiphoton excitation through wavelength mixing and serial block-face image acquisition. This approach provides organ-scale micrometric imaging of spectrally distinct fluorescent proteins and label-free nonlinear signals with constant micrometer-scale resolution and sub-micron channel registration over the entire imaged volume. We demonstrate tridimensional (3D) multicolor imaging over several cubic millimeters as well as brain-wide serial 2D multichannel imaging. We illustrate the strengths of this method through color-based 3D analysis of astrocyte morphology and contacts in the mouse cerebral cortex, tracing of individual pyramidal neurons within densely Brainbow-labeled tissue, and multiplexed whole-brain mapping of axonal projections labeled with spectrally distinct tracers. ChroMS will be an asset for multiscale and system-level studies in neuroscience and beyond.

[1] Laboratory for Optics and Biosciences, Ecole polytechnique, CNRS, INSERM, IP Paris, Palaiseau 91128, France. [2] Sorbonne Université, INSERM, CNRS, Institut de la Vision, 17 rue Moreau, Paris 75012, France. [3] Cold Spring Harbor Laboratory, Cold Spring Harbor 11724 NY, USA. [4] Department of Computer Science and Artificial Intelligence, University of the Basque Country, San Sebastian 20018, Spain. [5] IKERBASQUE, Basque Foundation for Science, Bilbao 48013, Spain. [6] Donostia International Physics Center (DIPC), San Sebastian 20018, Spain. [7] Center for Brain Science and Department of Molecular and Cellular Biology, Harvard University, Cambridge 02138 MA, USA. [8] Neurodegenerative Diseases Laboratory, Molecular Imaging Research Center, Institut de Biologie François Jacob, CEA, CNRS, Université Paris-Sud, Fontenay-aux-Roses 92265, France. These authors contributed equally: Solène Clavreul, Pierre Mahou. Correspondence and requests for materials should be addressed to J.L. (email: jean.livet@inserm.fr) or to E.B. (email: emmanuel.beaurepaire@polytechnique.edu)

Multicolor fluorescence microscopy has become a key enabling technology in biology by providing the means to spectrally resolve cells, organelles, or molecules within tissues and to analyze their interactions. Strategies combining multiple distinct fluorescent labels are increasingly used to study spatial relationships among cells and molecules and to encode parameters such as neuronal connectivity[1–9], cell lineage[10–15], cycling state[16,17], subtype identity[18], genotype[19,20], or signaling pathway activation[21]. Scaling up such approaches at the whole organ or tissue level would be a major step forward but is hindered by the lack of suitable large-volume multicolor microscopy methods. In recent years, serial block-face imaging, which relies on the automated, iterative alternation of imaging and microtome-based sectioning of a block of tissue, has been successfully transposed from electron microscopy to light microscopy[22–25]. This scheme has emerged as an effective means for generating data encompassing $mm^3$-to-$cm^3$ volumes of tissue at subcellular-resolution with either discrete or continuous sampling. One particularly active field of application is neuroscience, where block-face fluorescence imaging has opened the way to brain-wide mesoscale connectomics[23,26] and single neuron reconstruction efforts[26–29]. However, microtome-assisted microscopy methods developed so far are limited to single- or dual-color imaging[23,24,26,29]. This limitation is due to the general difficulty of exciting a manifold of fluorescent proteins. In addition, achieving tissue-scale multicolor microscopic imaging requires addressing chromatic aberrations and channel registration over large volumes. These obstacles made it so far very difficult to probe cell interactions or more generally to image multiplexed or combinatorial fluorescent signals with micrometer-scale precision in samples exceeding a few hundreds of microns in depth.

Here, we report on a method for volume multicolor and multicontrast microscopy with submicrometer registration of the image channels. Our approach, termed chromatic multiphoton serial (ChroMS) microscopy, relies on the integration of trichromatic two-photon excitation by wavelength mixing (WM)[30] with automated serial tissue sectioning. We show that ChroMS microscopy delivers multicolor imaging over >$mm^3$ volumes with constant micron-scale resolution and submicron channel co-registration, which sets new quality standards for large-scale multicolor microscopy. We demonstrate its performance for tridimensional imaging of mouse brains labeled by transgenic, electroporation-based or viral multicolor approaches. In addition to being perfectly suited for applications based on combinatorial fluorescence labeling, ChroMS microscopy also enables organ-wide imaging of label-free nonlinear signals such as third harmonic generation (THG) and coherent anti-Stokes Raman scattering (CARS). We illustrate its potential for high information-content three-dimensional (3D) imaging by demonstrating (i) analysis of astroglial cell morphology and contacts over several $mm^3$ of cerebral cortex, (ii) color-assisted tracing of tens of pyramidal neurons in a densely labeled, mm-thick cortical sample, and (iii) brain-wide color-based multiplexed mapping of axonal projection trajectories and interdigitation.

## Results

### Multicontrast organ-scale imaging with ChroMS microscopy.
In ChroMS microscopy, the sample (whole organ or large piece of tissue) is crosslinked with an embedding agarose block, and large-scale imaging is achieved by automatically alternating simultaneous multimodal acquisitions (four independent channels in the implementation used for this work), stage-based lateral mosaicking, and tissue sectioning with a vibrating-blade microtome. Simultaneous two-photon excitation in three distinct spectral bands is provided by the spatio-temporal overlap of two synchronized femtosecond infrared pulse trains at separate wavelengths $\lambda_1$ and $\lambda_2$ (Fig. 1a). For example, one can choose $\lambda_1$ = 850 and $\lambda_2$ = 1100 nm for optimal two-photon excitation of blue and red chromophores. Synchronization of the two pulses then gives access to two-color multiphoton excitation processes and in particular creates a 'virtual' two-photon excitation wavelength at $2/\left(\lambda_1^{-1} + \lambda_2^{-1}\right) \approx 950$ nm, suitable for excitation of green and/or yellow chromophores[30]. Tuning the power of the two pulses and their time delay enables to control the three excitation windows in a largely independent manner. This strategy provides efficient simultaneous two-photon excitation of several fluorescent proteins with emissions spanning the visible spectrum, as well as access to coherent label-free nonlinear signals such as third harmonic generation which reveals tissue structural details[31]. One specific feature of the WM excitation scheme is that it also enables CARS[32] imaging of a chosen vibrational mode along with detection of fluorescence and/or harmonic signals. A typical ChroMS acquisition involves sequential sectioning of the upper tissue block layer (typically 100 microns) and mosaic-based imaging of the remaining intact block (Supplementary Movie 1). The ChroMS setup can be programmed to acquire either 2D mosaics at discrete axial positions (e.g. every 100 μm) (brain-wide serial 2D or tomography mode, see Fig. 1a), continuous 3D volumetric images through the recording of partially overlapping image stacks (lossless continuous mode, see Fig. 1a), or combinations of both.

### Multicolor brain-wide anatomy with submicron resolution.
To establish the ability of ChroMS microscopy to deliver brain-wide diffraction-limited multicolor images, we recorded serial mosaics from brains of transgenic *Brainbow* mice (*CAG-Cytbow; Nestin-Cre*). Dense yet mostly mutually exclusive neural expression of Turquoise2/YFP/tdTomato is observed throughout the entire brain of these animals. Figure 1b, c, Supplementary Fig. 1 and Supplementary Movies 2–5 present a multicolor 3D whole-brain dataset consisting of 92 self-registered three-channel coronal planes with a lateral sampling of 0.55 μm and a discrete axial sampling of 100 μm. Each coronal plane, ranging from the olfactory bulb to the brainstem, constitutes a comprehensive submicron neuroanatomical map of labeled neurons and glial cells in which color contrast facilitates individual cell distinction. As shown in Fig. 1c, the resolution provided by ChroMS imaging enables to visualize fine neural processes such as Purkinje cell dendrites and mossy fiber axons in the cerebellum. ChroMS microscopy thus presents the sensitivity, spatial resolution, spectral discrimination and robustness necessary for micrometric multicolor imaging of native transgenic fluorescent labels over entire mouse brains. In the serial tomography mode illustrated here, it constitutes an effective approach for comprehensive anatomical mapping of different color labels with individual cell resolution at the whole-organ scale.

### Submicrometer color channel alignment over millimeter distances.
A unique feature of the ChroMS approach for large-volume multi-channel imaging is the intrinsic 3D submicron channel co-registration resulting from the excitation process, which makes it possible to assess and control chromatic aberrations over large volumes (Fig. 2). Indeed, green fluorescence signals obtained through wavelength mixing provide an intrinsic quality check: they are observed only if the $\lambda_1$ and $\lambda_2$ excitation point spread functions overlap in 3D with submicron precision, in turn ensuring that all signals originate from the same diffraction-limited volume. We mitigated chromatic aberrations as follows. We used second harmonic generation

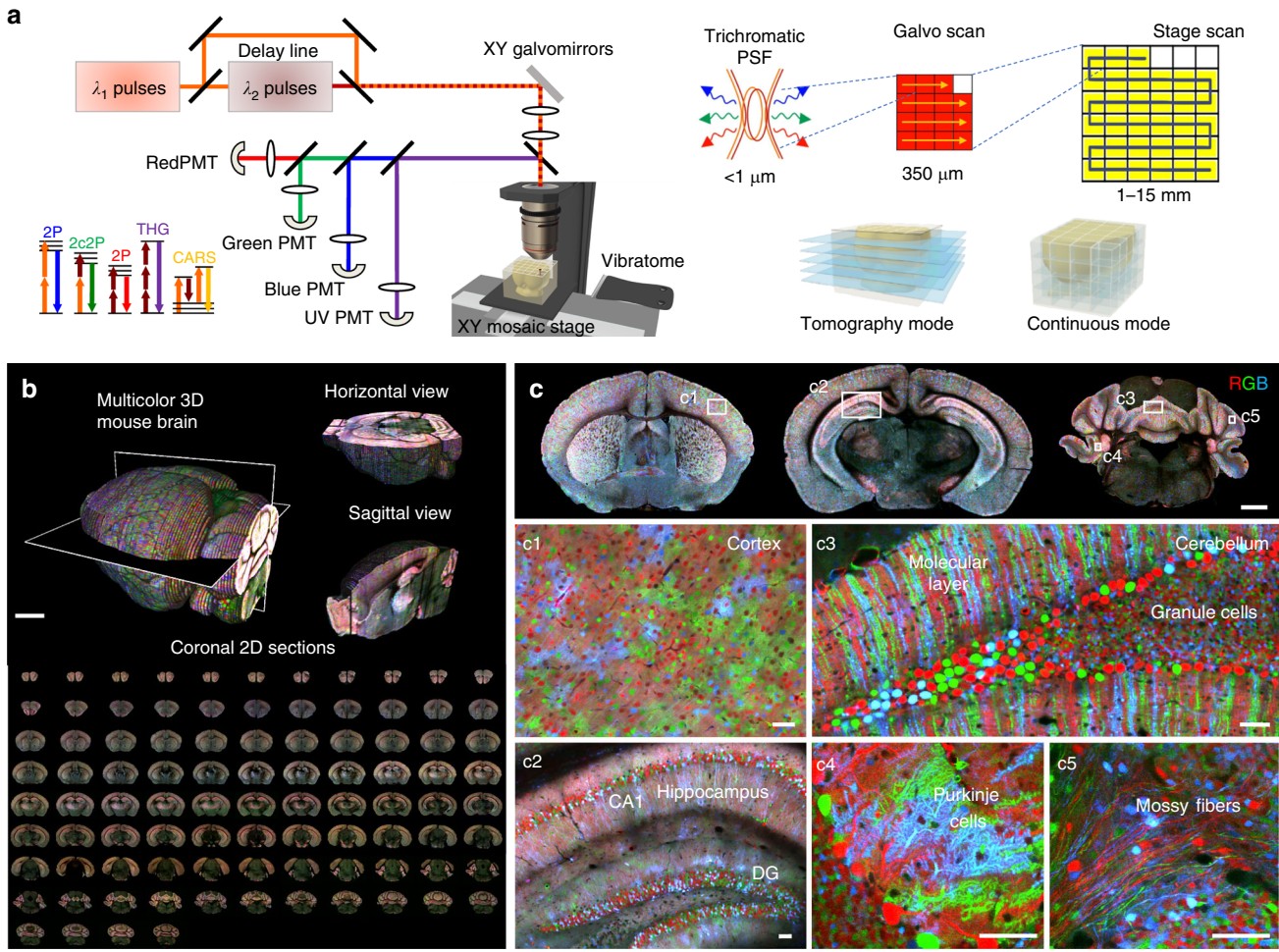

**Fig. 1** ChroMS microscopy principle and application for brain-wide multicolor imaging. **a** ChroMS imaging principle and setup (see Methods). **b** 3D views from a whole *CAG-Cytbow; Nestin-Cre* mouse brain ($10.5 \times 7.3 \times 9.2$ mm³) imaged with ChroMS microscopy. Scale bar: 1 mm. **c** 2D sub-cellular multicolor brain-wide anatomical maps from the brain in **b**. Upper panel displays representative coronal views. Middle and lower panel correspond to magnified crops from boxed regions in the coronal views, demonstrating sub-cellular lateral resolution enabling the visualization of dendritic and axonal processes. See also Supplementary Fig. 1 and Supplementary Movies 2–5. Scale bars: 1 mm (upper panel), 50 μm (middle and lower panels). DG: Dentate Gyrus

(SHG) from nanocrystals to map chromatic shifts across the field (Fig. 2a, b) and chose an excitation objective best suited for wavelength mixing as explained in ref [33]. We then optimized our scanning system to correct for lateral chromatic aberrations over a field of approximately 400 μm (see Methods). We inserted a telescope in each beam path prior to recombination in order to minimize their axial mismatch at focus. Overall, we obtained a color mismatch of less than 0.6 μm in all three dimensions over a field of view exceeding 350 μm (Fig. 2b). As illustrated in Fig. 2c, ratiometric imaging of neural tissue in such conditions enabled to access multiple resolvable colors resulting from combinatorial FP expression. Moreover, mosaicking and iterative sectioning preserved the nearly perfect matching between channels over cubic millimeters of imaged tissue (Fig. 2d). ChroMS microscopy therefore benefits from the combination of tissue sectioning and point-scanning two-photon imaging which ensures homogeneous volumetric resolution and robustness to depth-induced image degradation. These performances enabled us to visualize submicron structures such as neurites and astrocyte processes with multicolor precision (i.e. not affected by chromatic aberrations) at arbitrary depths (see Fig. 2d) for the first time. Finally, we point out that trichromatic single-shot excitation is advantageous in terms of imaging speed compared to sequential acquisition of

color channels, which is particularly relevant in the context of large volume acquisitions with point-scanning schemes[23,24,26].

**Multicolor 3D histology of uncleared neural tissue.** We then used ChroMS imaging to achieve continuous high-resolution 3D multicolor combinatorial imaging of uncleared brain tissue with constant resolution and contrast over the entire volume. Figure 3 and Supplementary Movies 6–8 show a $1.2 \times 2 \times 2$ mm³ portion of the cerebral cortex of an adult mouse brain continuously sampled with $0.4 \times 0.4 \times 1.5$ μm³ voxel size, labeled in a semi-dense manner by the MAGIC markers strategy[14]. This approach relying on electroporation of genome-integrative transgenes in embryonic neural progenitors efficiently labels with combinations of cyan, yellow, and red FPs the descent of these cells: glial cells, in particular astrocytes, distributed throughout the cortical wall, as well as pyramidal neurons, here restricted to layers 2–3. Post-acquisition linear unmixing was used to correct for the limited spectral bleed-through between channels due to the overlap of fluorescence emission spectra (Supplementary Fig. 2). In turn, this enabled colorimetric quantification of the trichromatic combinatorial labels expressed by the cells (Fig. 2c and Supplementary Fig. 3). As discussed in ref [14] and illustrated here, more than 20 different color combinations are readily resolved from

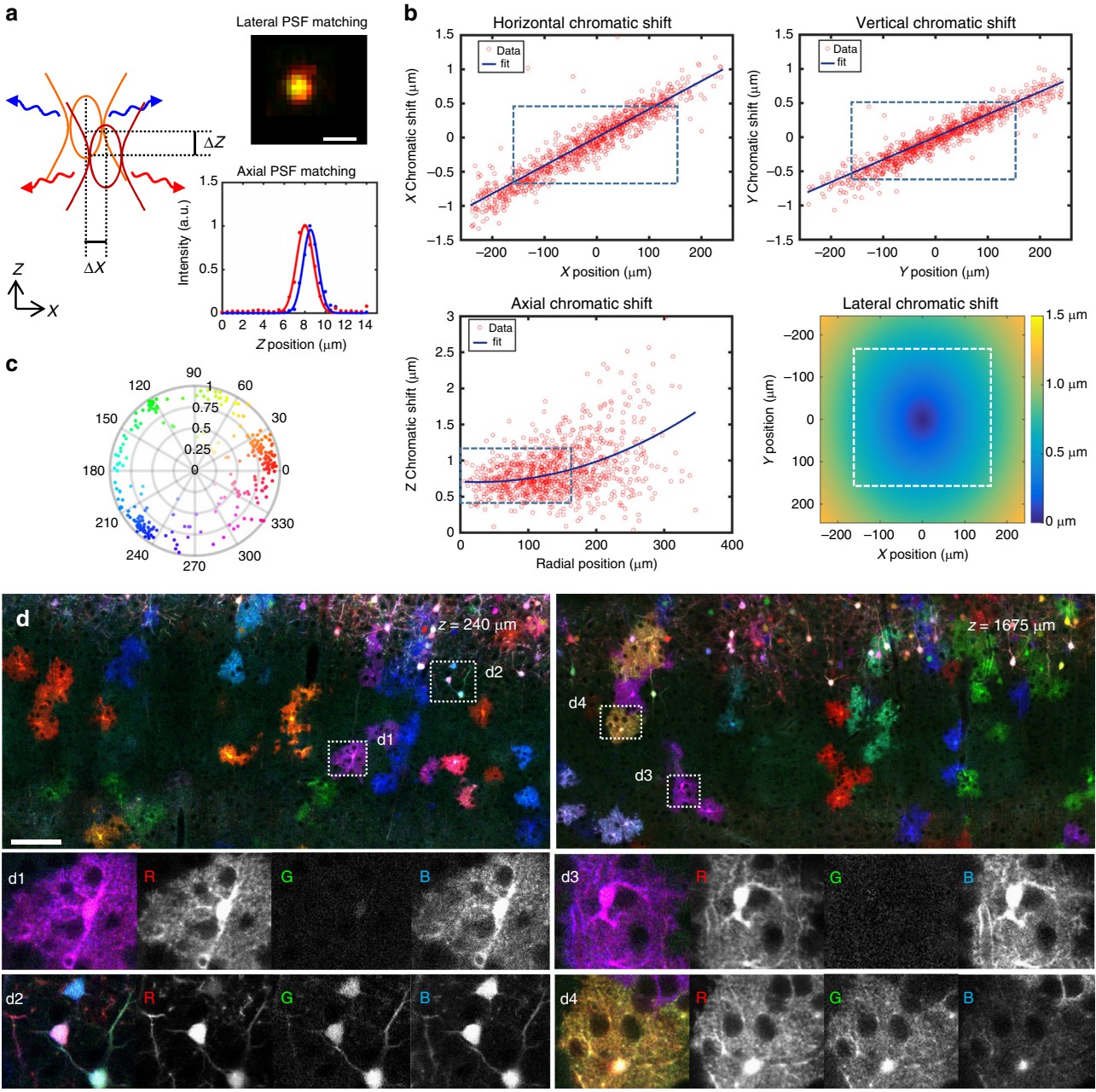

**Fig. 2** Submicrometer-scale channel registration over millimeter-scale volumes. ChroMS provides multicolor excitation in the 850–1100 nm range with <0.6 μm of chromatic shift in all three directions in the used field of view and at arbitrary depths. **a** Overlap of the two PSFs is required for wavelength mixing. Axial and lateral chromatic shifts are measured using SHG nanocrystals. Top right: coalignment of the foci at the center of the field of view (fov). Bottom right: axial PSFs corresponding to the two beams measured were axially matched using telescopes placed on each beam path. **b** Characterization of chromatic aberrations across the fov for the 850 nm/1100 nm wavelength combination. Top: Chromatic lateral shifts $\Delta X$ and $\Delta Y$ as a function of the position in the fov. Measurements were performed on $n = 1012$ KTP nanocrystals across the fov and data fitted using an affine model. Bottom left: Chromatic axial shift $\Delta Z$ as a function of the radial position in the fov. Measurements were performed on $n = 1037$ KTP nanocrystals across the fov and data fitted using a polynomial model. Bottom right: Lateral chromatic shift mapped across the fov. Lateral chromatic shift is defined as $(\Delta X^2 + \Delta X^2)^{0.5}$ with $\Delta X$ and $\Delta Y$ the corresponding fitted horizontal and vertical chromatic shifts. Dashed boxes outline the effective fov used for ChroMS imaging experiments. **c** Hue-Saturation polar plot representing the distribution of color combinations expressed in a portion of cortex labeled with the MAGIC markers strategy after spectral unmixing. Multiple color combinations (>20) spanning the HSV color space can be distinguished[14]. **d** XY representative multicolor images from a MAGIC markers-labeled dataset acquired with ChroMS microscopy with 0.4 μm × 0.4 μm × 1.5 μm voxel size. Insets demonstrate channel co-registration at arbitrary depths within the dataset. Individual channels (R, G, B) are shown in gray scale. Cell structures (astrocyte domains and main processes, cell bodies, axons) can be visualized with multicolor precision (i.e. not affected by chromatic aberrations) and discriminated with a quantitative color ratio. Scale bar: 100 μm

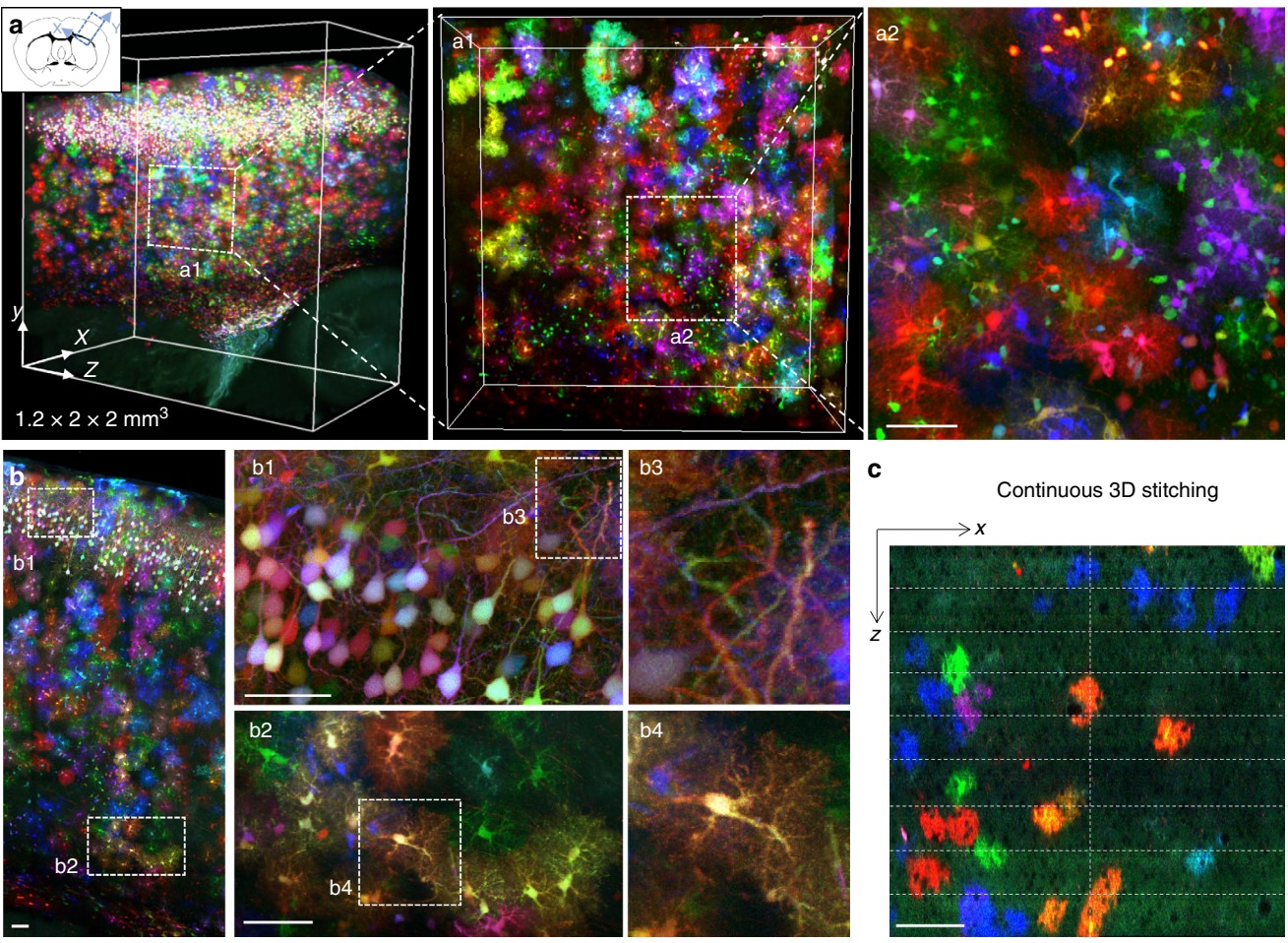

**Fig. 3** Continuous 3D multicolor imaging of mouse cortical tissue. **a** Multicolor 3D micrometric imaging of a 1.2 × 2 × 2 mm³ volume of a P65 mouse cerebral cortex with ChroMS microscopy, showing multicolor labeling with MAGIC markers. Scale bar: 50 μm. **b** Maximum intensity projection over 500 μm and XY crops showing high resolution imaging of axons and astrocyte processes with multicolor precision. Scale bars: 200 μm (right panel) and 50 μm (middle panel). **c** XZ view of nine successively imaged blocks stitched in 3D, showing the continuity of the dataset in depth. Cellular structures are recovered from one block to another with no tissue loss. Scale bar: 100 μm. See also Supplementary Movies 6–8

such multichannel data. We finally performed a 3D stitching of the multi-channel blocks (see Methods) and obtained a continuous dataset giving access to high-resolution 3D morphology of labeled cells over the whole imaged cortical volume (Fig. 3). In particular, thousands of protoplasmic astrocytes were imaged in the cortical wall with micrometer-scale resolution. The cell bodies and main cytoplasmic processes of these cells as well as the tridimensional domain of neural tissue occupied by their fine processes[34] were resolved throughout the entire volume (Fig. 3a, b). This dataset also demonstrated multicolor imaging of the densely labeled neurons located in upper cortical layers, enabling to resolve their neurites (Fig. 3b).

**Quantitative 3D cell morphometry of cortical astrocytes.** Astrocytes, the main glial cell type of the brain, form an uninterrupted cellular network throughout the neuropil[34]. Studying the organization of this network faces two difficulties: singling-out individual astrocytes from their neighbors and imaging these cells in their entirety. The multicolor labeling of astrocytes in the continuous tridimensional data presented above provided a unique opportunity to study their layout and contacts without biases introduced by tissue sectioning (Fig. 4).

We first derived a reconstruction of the astroglial network over a depth of 1 mm from a volume containing 1055 labeled

astrocytes which we all positioned in 3D (Fig. 4a, b and Supplementary Fig. 4). These astrocytes often formed clusters of juxtaposed cells expressing an identical color ($n = 261$, Fig. 3a), as expected from clonally related cells generated by local proliferation[35]. The finely resolved ChroMS images made it possible to visualize the somata of the cells engulfed within the domain of the labeled astrocytes (mostly neuronal, negatively contrasted in the images): we pointed 12,265 such contacting cell bodies (Fig. 4a, b, Supplementary Fig. 4). By computing the average soma-to-astrocyte ratio in each cluster, we inferred that each mouse cortical astrocyte was in contact with an average of $12.8 \pm 4.6$ (mean ± s.d, $n = 248$, median = 12.3) cells (Fig. 4c). Interestingly, we observed that this parameter displayed significant variations across cortical layers (Fig. 4c): layers 4 and 6 astrocytes contacted more cells in average than the general distribution, $17.2 \pm 3.6$ ($n = 18$, median = 17.8) and $17.1 \pm 4.9$ ($n = 35$, median = 16), respectively. Layer 5a and 5b astrocytes contacted significantly less cells than the former categories with an average of $8.9 \pm 2.7$ ($n = 20$, median = 8.6) and $9.6 \pm 2.6$ ($n = 17$, median = 9.5) contacted cells for layer 5a and 5b astrocytes, respectively.

We then performed 3D morphological analyses of individual astrocytes, taking advantage of isolated or color-segregated cells present in the dataset. We could in this manner analyze 130 astrocytes within a single brain. We segmented the domains (a.k. a. territory) covered by the processes of these astrocytes within

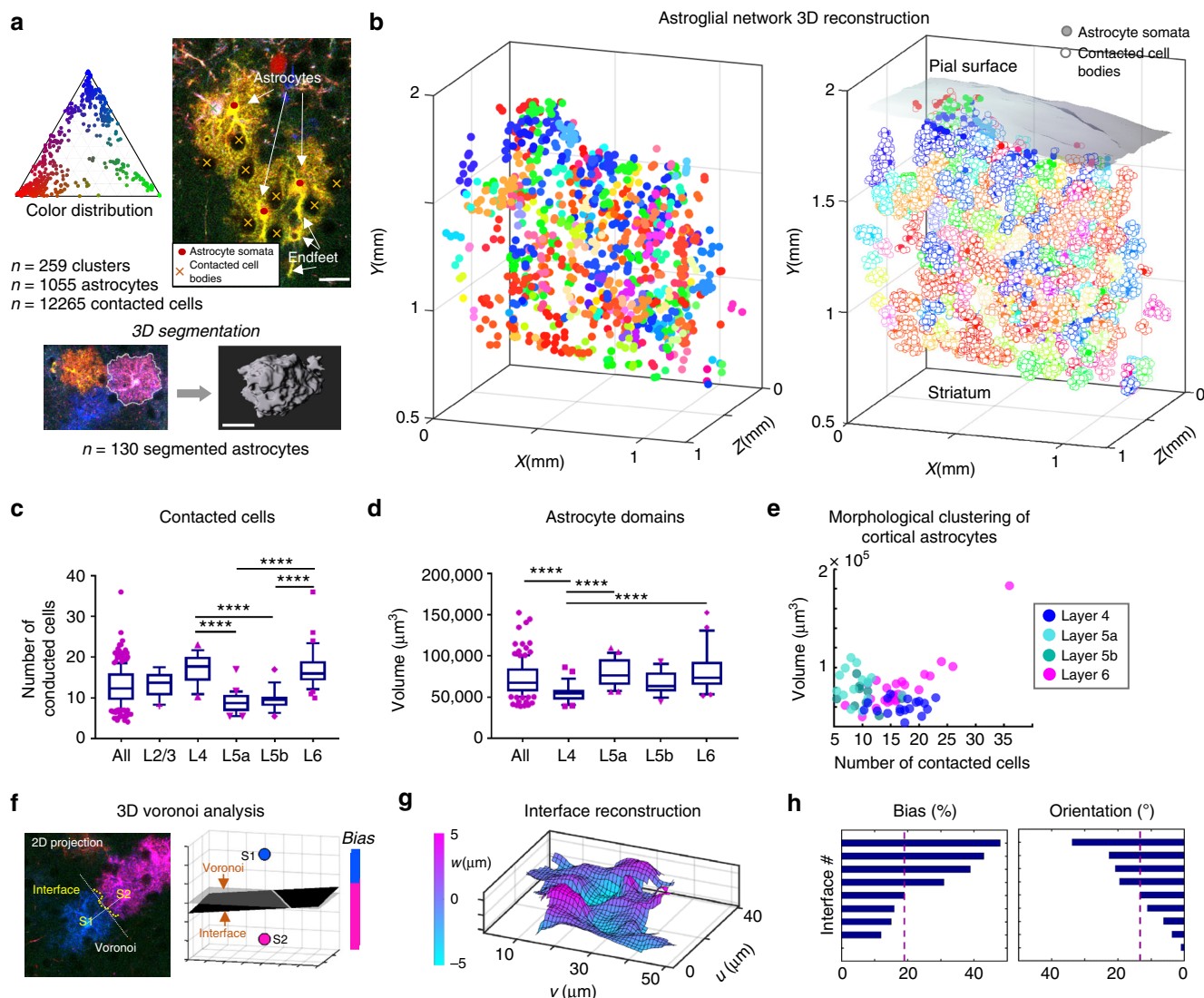

**Fig. 4** 3D analysis of cortical astrocyte morphology and heterogeneity across layers. **a** Colorimetric and morphological analyses performed on labeled astrocyte in the multicolor dataset shown in Fig. 3. Top left, ternary plot of the color distribution of astrocyte clusters present in the dataset (ensemble of astrocytes expressing a same color combination and continuously in contact with each other). Top right: positioning of astrocyte somata from a cluster and the cells they contact (visible as negatively contrasted). Bottom: 3D segmentation of astrocyte domains. **b** 3D plots showing labeled astrocytes (color-filled) without (left) or with (right) their contacted cells (white-filled markers) in a 1.2 × 2 × 1 mm³ subvolume of the multicolor dataset shown in Fig. 3. The 3D positions of 1055 glial cells forming 261 color clusters and 12,265 contacting neurons have been extracted in the entire dataset. **c**, **d** Number of contacted cells per astrocyte (**c**) and 3D segmented volumes of astrocyte domains (**d**) as a function of cortical layers. Data are presented as box-and-whisker plots, central mark representing the median value, bottom and top box edges indicating the 25th and 75th percentiles, whiskers extending to 10 and 90% of min and max values. Non-parametric one-way statistical test (Kruskal–Wallis) followed by post hoc multiple comparison tests (Dunn) have been performed. **** indicates adjusted p-value <0.0001. **e** Domain volume as a function of the number of contacted cells for individual astrocytes located in cortical layers 4–6. **f–h** Morphological analysis of astrocyte-astrocyte contacts. The interface is reconstructed in 3D and compared to the median position given by a Voronoi tessellation. The proportion of interface on either side of the median plane is computed (bias %) as well as the interface mean orientation (°) relative to the median plane. See also Supplementary Figs. 5 and 6

the imaged cortical portion to measure their volume (Fig. 4a). We derived from the 3D segmentation an average cortical astrocyte domain volume of $7.3 \times 10^4$ μm³ $\pm 2.2 \times 10^4$ (mean ± s.d, $n = 130$) with a median value at $6.8 \times 10^4$ μm³, close to that measured in other studies in vivo ($6.4 \times 10^4$ μm³ $\pm 4314$ in ref. [36]), but divergent from previous reports based on sucrose[37] or CUBIC-based[38] clearing media, known to modify tissue volume. Using our data, we also investigated the variation of astrocyte territorial volumes across cortical layers (Fig. 4d). We focused our analysis on singled-out astrocytes in deep cortical layers (layers 4–6). This analysis revealed a significant volume difference (adjusted p-value

<0.0001) between layer 4 astrocytes ($5.5 \times 10^4$ μm³ $\pm 1.2 \times 10^4$, $n = 29$, median = $5.5 \times 10^4$ μm³) and astrocytes located in projection layer 5a ($8.1 \times 10^4$ μm³ $\pm 1.7 \times 10^4$, $n = 22$, median = $7.7 \times 10^4$ μm³), as well as a significant volume difference between layer 4 astrocytes and deep layer 6 astrocytes ($8 \times 10^4$ μm³ $\pm 2.2 \times 10^4$, $n = 43$, median = $7.4 \times 10^4$ μm³). We next evaluated at the single cell level whether this layer-dependent volume variation was correlated with the soma-to-astrocyte ratio (Fig. 4e), and found no significant correlation (correlation coefficient $r = 0.265$, $n = 67$). However, the analysis revealed different distribution patterns according to cortical layers: layer 5a astrocytes occupied

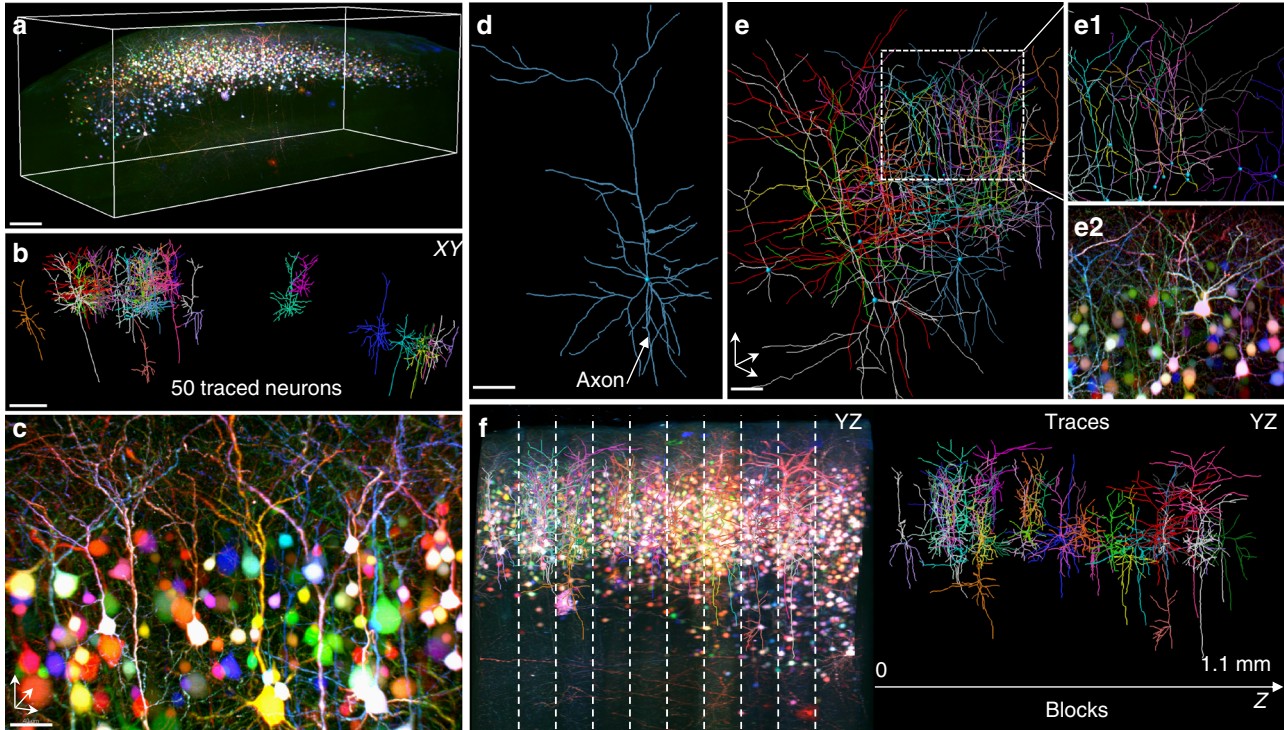

**Fig. 5** Dense tracing of multicolor-labeled cortical pyramidal neurons. **a** 3D view of a $2.6 \times 0.9 \times 1.1$ mm$^3$ portion of cerebral cortex electroporated at E15 with a *CAG-Cytbow* transgene, yielding dense multicolor labeling of layer 2/3 pyramidal neurons. Scale bar: 200 µm. **b** XY projection of 50 reconstructed neurons within the volume displayed in (a). Scale bar: 150 µm. **c** 3D view from the volume in **a** showing color segregation of tufted apical dendrites from individual adjacent neurons. Scale bar: 40 µm. **d** Example of 3D reconstruction of a pyramidal neuron. Scale bar: 50 µm. **e** 3D view showing multiplexed neuron tracing in a densely labeled area. Insets (**e**1) and (**e**2) show, respectively, a zoom of the boxed area in **e** and the corresponding color image. Scale bar: 50 µm. **f** 3D neuron tracing across serially acquired image blocks. Left: YZ projection of the 3D volume represented in **a**. Dotted vertical lines correspond to junctions between blocks. Right: YZ view of the corresponding neuron traces. See also Supplementary Movies 9 and 10

larger volumes while contacting fewer cells; layer 4 astrocytes occupied small volumes and contact more cells; layer 6 astrocyte volume increased with the number of contacted cells (correlation coefficient $r = 0.705$, $n = 25$).

Additionally, color contrasts among labeled astrocytes provided a unique opportunity to study how their domains partition the neuropil. We implemented a workflow to analyze astrocyte–astrocyte interfaces in 3D on the same dataset, taking advantage of the color contrast to extract the domain frontiers between pairs of neighboring astrocytes labeled with distinct FP combinations (Fig. 4f–h, Supplementary Figs. 5 and 6). For each dual-color astrocyte pair ($n = 9$), we reconstructed in 3D the surface of contact and analyzed its position and orientation relative to the cell–cell median plane given by a Voronoi tessellation based on nuclei positions. This analysis revealed frequent imbalance in the positioning of the limits of astrocyte cytoplasmic domains with respect to the median plane (mean bias parameter 75.2% ± 15.2 [s.d.]), indicating that the tiling of the cerebral cortex by astroglial cells deviates from an equiparted Voronoi geometry.

Together, these results show that astrocytes present significant heterogeneity in the volume of their domains and in their local partitioning of the neural tissue, suggesting that they develop in a plastic rather than fixed manner and may adapt to both their local environment and neighbors during cortical development and maturation. Our findings illustrate the unique strengths of continuous multicolor 3D imaging with ChroMS microscopy for unbiased quantitative 3D analysis of cell morphology and contacts within large tissue volumes. Specific methodological advantages of ChroMS for that purpose include the following: (i) the color contrast increases the throughput of individual cell

analysis for a given sample by alleviating the need to use very sparse labeling; (ii) it enables one to analyze contacts between cells labeled with different colors; (iii) the serial multiphoton approach suppresses the orientation and size artefacts due to section registration or tissue shrinking/swelling encountered respectively in histology and clearing techniques.

**Color-based neuron tracing over extended volumes**. We then explored the possibilities provided by ChroMS imaging for multiplex tracing of neuronal processes within a crowded environment (Fig. 5). While state-of-the-art high-speed serial tomography platforms make it possible to trace entire neuronal morphologies across the brain[26,29], the throughput of such approaches remains limited to a few neurons per brain, labeled in a highly sparse manner. Brainbow color labels offer a way to alleviate this limitation, but their use for multiplexed neuron tracing has so far been limited, essentially because of the difficulty to generate large-scale chromatically corrected multicolor datasets. We here show that ChroMS imaging enables color-based multiplex neuron tracing within large tissue volumes (Fig. 5a–f, Supplementary Movies 9 and 10). We imaged a portion of P21 mouse cortex electroporated with a *CAG-Cytbow* transgene[14] at E15, a condition resulting in relatively dense labeling of layer 2/3 pyramidal excitatory neurons (Fig. 5a, c, Supplementary Movie 9). A $2.6 \times 2 \times 1.1$ mm$^3$ multicolor volume was acquired using the continuous ChroMS imaging mode with a $0.54 \times 0.54 \times 1.5$ µm$^3$ voxel size over a depth of 1.1 mm, corresponding to the assembly of ten acquired 3D blocks. Within this volume, we traced 50 individual pyramidal neurons (Fig. 5b–d, Supplementary Movie 10), including crowded ensembles with neighboring

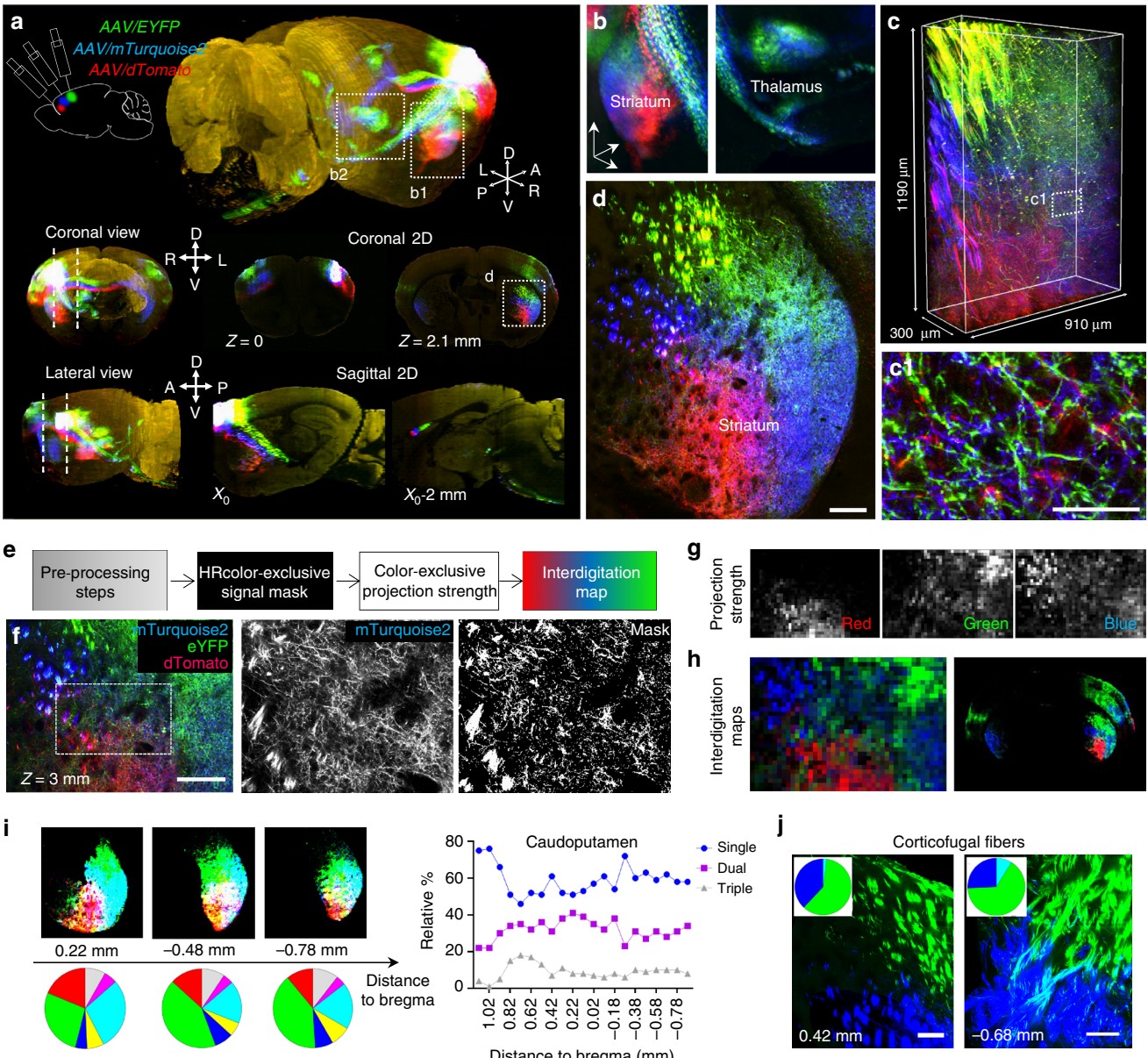

**Fig. 6** Brain-wide multiplexed projection mapping with ChroMS microscopy. **a** Brain-wide mapping of neural projections using tricolor AAV anterograde labeling and ChroMS imaging. 3D, coronal and sagittal views of the 3 labeled projections. See also Supplementary Movies 11–14. **b** 3D views extracted from the multicolor volume showing tridimensional topography patterns in target regions (striatum, thalamus). **c** Continuous high-resolution 910 × 1190 × 300 μm³ volume acquired with 0.4 × 0.4 × 1.5 μm³ voxel size in the striatum, within the brain-wide dataset. (**c1**) Magnification of the boxed region in **c** showing intermingled axonal processes. Scale bar: 30 μm. **d** Magnification of the region boxed in **a** showing topographic arrangement of labeled projections in the striatum. Scale bar: 200 μm. **e–h** Multiplexed quantitative projection analysis. The general workflow is summarized in **e**. See also Supplementary Figs. 7–10. High-resolution binary signal masks (**g**) are computed for each spectrally unmixed channel to separate fluorescence signal from background. For each channel mask, pixels detected in other channels are removed to generate color exclusive binary masks. The signal ratio over non-overlapping blocks of 30 × 30 pixels (24 × 24 μm²) is then computed to generate three exclusive projection strength maps (**g**). Red, green, and blue projection strength maps are merged into quantitative color maps of interdigitation (**h**). Scale bar (**f**): 200 μm. **i** Quantitative projection analysis in the caudoputamen (CP). Left: Three representative interdigitation maps at different rostro-caudal positions throughout the CP and corresponding interdigitation diagrams. Diagram color code: red, green, and blue correspond to single red, green, and blue projections, respectively; yellow, cyan, and magenta correspond to dual red+green, green+blue, and red+blue projections, respectively; gray corresponds to interdigitation of the three projections. Right: relative proportion of single, dual, and triple projections throughout the caudoputamen. See also Methods, Supplementary Fig. 11 and Supplementary Movie 14. **j** Analysis of the topography of striato-pallidal fiber arrangement reveals segregated parallel pathways throughout the striatum (left) followed by partial fiber interdigitation when passing through the external pallidus (right). Scale bars: 100 μm. See also Supplementary Fig. 12

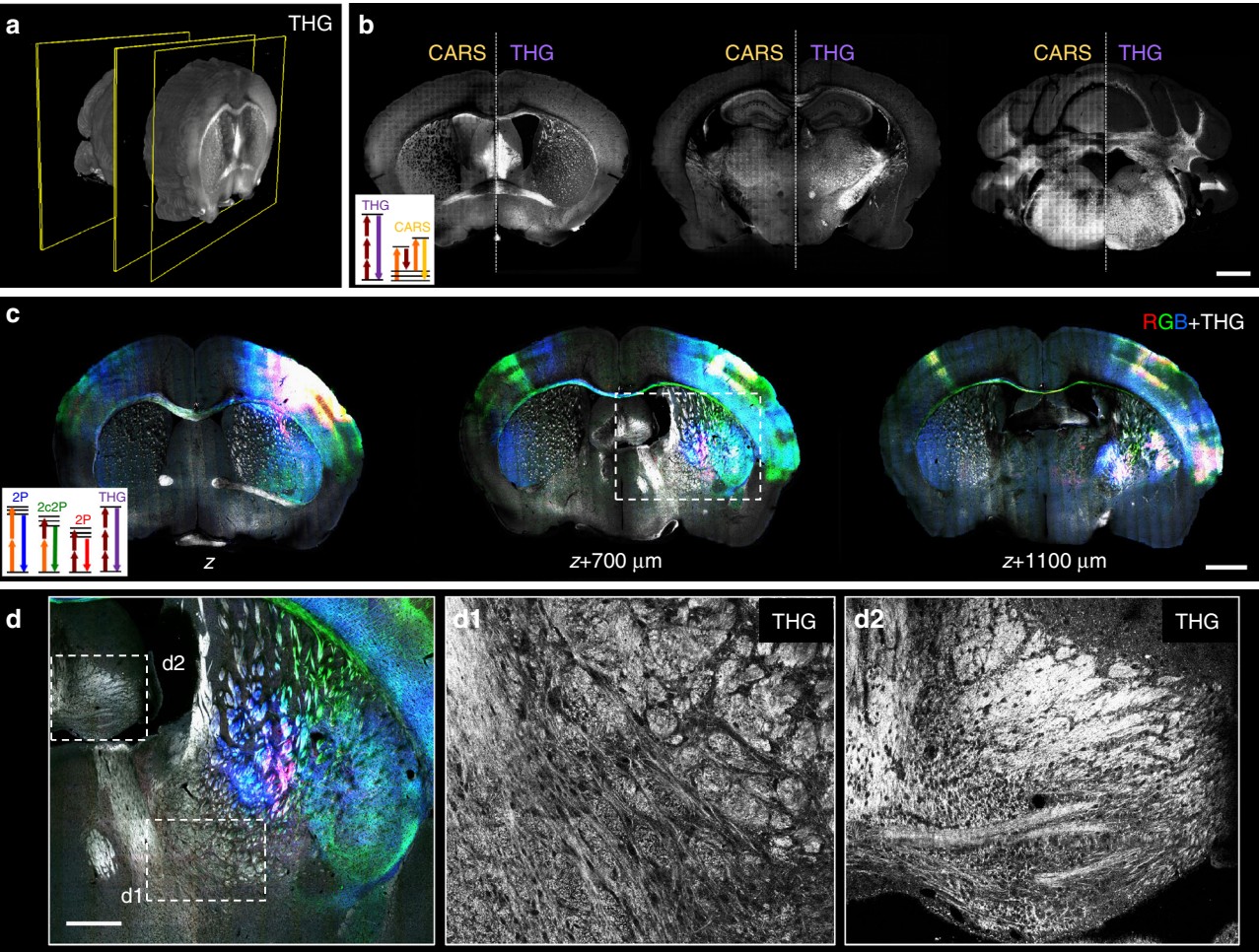

**Fig. 7** Brain-wide multimodal imaging using endogenous nonlinear contrasts. **a**, **b** Label-free THG-CARS serial imaging. **a** Coronal 3D view from the brain-wide THG channel. **b** Representative coronal sections showing combined THG/CARS images acquired with ChroMS imaging. Excitation wavelengths set to 832/1090 nm match the CH2 stretching vibrational band at 2845 cm$^{-1}$ and produce CARS at 673 nm. THG from the 1090 nm beam is detected at 363 nm. Scale bar: 1 mm. **c**, **d** Multicolor+THG coronal sections extracted from a 8 × 11 × 5 mm$^3$ volume using 840/1100 nm excitation. Three AAV anterograde tracers (DsRed Express, YFP, and mCerulean) were injected in motor and somato-sensory cortex. **c** Overlay of multicolor (RGB) and THG (grays) 2D coronal images acquired simultaneously. Scale bar: 1 mm. **d** Magnification of the boxed area in **c** showing multicolor-labeled neural projections along with morphological landmarks. THG signals highlight cytoarchitecture and lipid-rich structures such as myelinated axons (**d**1–**d**2). Scale bar: 500 μm

somas and intermingled dendrites (Fig. 5e). Compact groups of cells could be unambiguously traced thanks to the high color contrast provided by the acquisition scheme, enabling to disentangle their dendrites. This confirmed the scalability of ChroMS imaging for tracing multiple long neurites extending across slicing blocks (Fig. 5f).

**Multiplexed brain-wide mapping of neuronal projections.** Interrogating the anatomical convergence and segregation of axonal projections from different sets of neurons across large distances is fundamental for gaining insight into the functional organization of the nervous system. Until now, structural and functional studies of interacting projections have mostly relied on registration of different projection traces to a common framework[24,39]. Using distinct color labels can more precisely inform on the anatomical relationships between different tracts in the same brain volume[5–7,29], but a convenient approach to map these labels in 3 dimensions has been lacking. Here using ChroMS microscopy, we achieved one-shot whole-brain tricolor projection tract imaging on mouse brains injected with three distinct anterograde AAV tracers (AAV-tdTomato, AAV-EYFP

and AAV-mTurquoise2) in three discrete locations of the sensorimotor cortex (Fig. 6, Supplementary Figs. 7–10, Supplementary Movies 11–14). We used the whole-brain serial 2D mode of ChroMS microscopy (Fig. 6a, b), combined with high-resolution (0.4 μm × 0.4 μm × 1.5 μm) continuous-mode imaging of a 910 × 1190 × 300 μm$^3$ sub-volume within the striatum (Fig. 6c). This hybrid acquisition scheme enabled us to obtain both a global map of the labeled projections and a detailed view of their axonal components in a given target region. Multicolor imaging of the three distinctly labeled neuronal populations within the same brain provided immediate access to the relative arrangement of their projections, enabling to directly assess their topographic arrangement and spatial overlap. The dataset presented here reveals large-scale topography patterns, such as conserved topographic order within the callosal tract and striatum and a core-shell topography in the thalamus, which corroborate previous studies[6,7] (Fig. 6a–d). In addition, a unique feature of simultaneous multicolor projection mapping is that it enables direct analysis of the local fine scale interdigitation of axonal projections from remote sites. For instance, while axons originating from the three labeled cortical areas followed globally distinct trajectories in the striatum (Fig. 6b–d), the high resolution multicolor images

revealed local intermingling at the tens of microns scale (Fig. 6c). To quantify these observations over the entire dataset, we developed an image analysis workflow for high-resolution segmentation (derived from ref. [24]) and multiplexed analysis of color-labeled projections from initially segregated regions (Fig. 6e–j, Supplementary Figs. 7–12). This enabled us to generate quantitative maps of projection strength (Fig. 6h, Supplementary Fig. 8) which provided an estimate of the number of labeled axonal branches within a given surface (or super-pixel, set here to 24 μm × 24 μm). By overlapping projection strength maps, we created interdigitation maps that revealed the respective contribution of the labeled projections in each super-pixel, and in which strong vs. weak projections could be differentially highlighted (Supplementary Fig. 9). Applying this workflow in the striatum enabled to visualize the relative contribution of the cortico-striatal projections defined by the three AAV labels and to quantitatively probe their interdigitation in this target region (Fig. 6i, j, Supplementary Figs. 11 and 12, Supplementary Movies 11–14). This analysis confirmed that each of the three projection components occupied a subdomain of the Caudo-Putamen (CP) according to the known topography of this structure, but also indicated that the proportion of the CP occupied by one, two, or three of the projections remained within similar values at each rostro-caudal level (Fig. 6i, Supplementary Fig. 11). We also used the same workflow to quantitatively characterize the topography of cortical fibers passing throughout the striatum. While fibers labeled in blue and green, corresponding to the intermediate and most caudal injections, traveled throughout the striatum in a strictly segregated manner, they started to interdigitate upon joining the internal capsule (Fig. 6j, Supplementary Fig. 12). Thus, initially segregated projections progressively intermix at the fine scale level as they course through the striatum, implying an alteration of their topographic arrangement. This analysis illustrates how our method enables precise interrogation of both anatomical segregation and convergence of color-labeled axon tracts within a brain area. ChroMS imaging of multiple distinctly labeled neuronal projections therefore provides an effective means to assess their relative anatomical relationships at multiple scales within a single brain.

**Label-free brain-wide micrometric morphological imaging**. Finally, the ChroMS system can be used to generate label-free images based on endogenous nonlinear optical signals (Fig. 7). Coherent contrast modalities such as THG and CARS tuned to $CH_2$ stretching mode have been shown to reveal morphology and myelinated fibers in brain tissue[40–42] but cannot be used for organ-scale imaging with standard microscope configurations. Although both modalities highlight lipidic structures in brain tissue, they rely on different contrast mechanisms and therefore can provide complementary information on sample microstructure[43]. Figure 7a, b presents label-free THG-CARS imaging obtained using mixed 832/1090 nm excitation, recorded here on a brain-wide scale. Figure 7c, d also illustrates trichromatic fluorescence imaging of viral-labeled axonal projections recorded simultaneously with label-free THG signals channel using 850/1100 nm excitation. This configuration is particularly informative, as the THG image complements the color-based projections with a rich morphological background, highlighting myelinated axons, fiber tracts, and negatively contrasting cell bodies.

## Discussion

We present here a multicolor/multi-contrast tissue-scale microscopy method which bypasses the spectral limitation common to current large volume imaging techniques, and in particular

expands the field of applications of serial two-photon microscopy[23,44].

ChroMS microscopy achieves efficient one-shot excitation in three distinct spectral bands, enabling multicolor imaging of minimally processed tissue with sub-micron channel alignment and resolution preserved over cubic millimeter volumes. The integration of WM-excitation and serial block-face imaging solves two general problems encountered in tissue imaging: chromatic aberrations (inherent to all multicolor microscopy approaches) and inhomogeneous image quality across volumes. ChroMS microscopy can generate brain-wide atlas-like color datasets with subcellular resolution as well as continuous 3D datasets of supra-millimetric size with submicron multicolor precision and diffraction-limited axial resolution.

Although clearing techniques (e.g. CLARITY[45], CUBIC[46], or iDISCO[47]) combined with light-sheet microscopy have constituted major breakthroughs in large-volume and organ-scale tissue imaging with single-cell resolution, none of the plane illumination techniques available thus far have been able to deliver multicolor images with high contrast in all channels, together with consistent and homogeneous submicron resolution across large volumes. Multicolor high-resolution imaging of genetically labeled samples – especially red/green/blue combinations – using serial blockface imaging methods also represented a technical challenge so far. Indeed, previously reported blockface methods either relied on a single two-photon excitation wavelength which restricted contrast to dual-color red/green imaging[7,26], or involved tissue processing steps causing fluorescence quenching/recovery[29] which were not compatible with a large palette of fluorescent proteins. This bottleneck made it especially difficult to image combinatorially labeled tissues over depths exceeding a few hundreds of microns. In particular, the growing toolbox of multicolor labeling approaches such as Brainbow[1], Confetti[48], MADM[10], MAGIC markers[14], Startrack[15], IfgMosaics[19], etc., has not yet fully benefited from large volume microscopy methods actively developed in recent years. As we have shown, the ChroMS approach overcomes these limitations. Another advantage of the minimal tissue processing associated with ChroMS imaging relative to the above approaches is that it limits the possibility of introducing tissue distortions. In fact, our astrocyte territory measurements were close to those formerly obtained in vivo ($6.4 \times 10^4$ μm³ ± 4314 in[36]), but less consistent with two studies performed on clarified tissue slices ($2.4 \pm 10^4$ μm³ in ref. [37], >9 $\pm 10^4$ μm³ in ref. [38]). Indeed it has been reported that dehydration-based methods[37] induce tissue shrinkage and that hyperhydration-based methods[38] generally induce tissue expansion. Thus ChroMS imaging appears as a method of choice for performing morphometric studies with reduced biases.

We note that one limitation of the ChroMS system described here is the image acquisition time, as imaging an entire mouse brain with discrete axial sampling currently takes 2–3 days. This caveat, however, is neither unbreakable nor truly limiting: imaging speed in a point-scanning approach is essentially limited by the fluorescence flux, which sets the minimal pixel dwell time. As shown in the context of monochrome serial two-photon imaging, acquisition can be significantly accelerated by using a fast scanning scheme in combination with strongly labeled samples such as AAV-labeled tissues, and automated contour detection[26]. With such optimizations, Economo et al reported 7–10 acquisition days for imaging a mouse brain with continuous micrometer-scale sampling[26]. Furthermore, there is a most pressing need, common to all large volume microscopic techniques, to establish efficient workflows to handle and analyze large datasets, as image analysis time usually exceeds by far the acquisition time. Despite

its speed limitation, the current implementation of ChroMS delivers previously unachievable performance for multicolor imaging.

Importantly, all applications illustrated here in the brain can also be envisioned for organ-scale studies in other tissues. The continuous imaging modality enables high-resolution anatomical reconstruction and analyses of color labeling patterns over extended volumes (in essence, 3D histology). Combined with the expanding transgenic multicolor toolbox available to label cells[1,2,4,18], report specific cellular events and molecular signals[3,11,14,17,21,49] or analyze the effect of experimental perturbations[19,20], this work paves the way to large scale mapping of cellular networks organization, development[14,50] and function. In the brain-wide tomography mode, ChroMS imaging will facilitate mesoscale connectomics efforts, using color labels to resolve subgroups of neurons within neural projections or map putative convergence from distant brain regions[5,7,51,52]. Strategies targeting specific cell-type or projections based on combined genetic drivers, reporters and viral vectors[53,54] further expand the spectrum of our method's foreseen applications. In addition, ChroMS microscopy provides label-free nonlinear contrasts i.e. microstructural and chemical information[40,41] so-far inaccessible at the organ scale. Multimodal imaging relying on multicolor fluorescence and harmonics or vibrational signals may be used to develop novel pathophysiological screening modalities. ChroMS will therefore be an asset for multiscale and system-level studies in neuroscience and beyond.

## Methods

**Multicolor transgenic mice**. Mice were housed in a 12 h light/12 h dark cycle with free access to food, and animal procedures were carried out in accordance with institutional guidelines. Animal protocols were approved by the Charles Darwin animal experimentation ethical board (CEEACD/No 5). We used transgenic mice broadly expressing a *CAG-Cytbow* transgene[14], modified to achieve random expression of mTurquoise2[55], mEYFP[56] or tdTomato[57] from the broadly active *CAG* promoter following Cre recombination. These mice were crossed with *Nestin-Cre (B6.Cg-Tg(Nes-cre)1Kln)* animals[58] and their offspring analyzed at P29. This strategy triggers multicolor labels in Nestin+progenitors and their descendants, including both neuronal and glial cells.

**Labeling of cortical astrocytes and upper layer pyramidal neurons**. Cortical astroglial cells were labeled by targeting their progenitors in the dorsal tele-ncephalon with the MAGIC markers strategy[14]. A glass micropipette was used to inject into the lateral ventricle of E15 Swiss embryos 1 µl of DNA mix containing the three following plasmids: *PBCAG-Cytbow* (2 µg µl⁻¹), in which PiggyBac transposition endfeet flanking a *CAG-Cytbow* cassette enable genomic integration, *CAG-hyPBase* (1 µg µl⁻¹), expressing the corresponding transposase[59], and *CAG-seCre* (0.2 µg µl⁻¹) which expresses a self-excisable Cre recombinase. Four 35 V pulses of 50 ms were applied with a CUY21EDIT electroporator (NepaGene) and 3 mm diameter Tweezertrodes (Sonidel Limited) positioned to target the dorsal wall of the lateral ventricle. Electroporated animals were allowed to develop until analysis at P65. This procedure efficiently labels with FP combinations the cells originating from the electroporated cortical progenitors, i.e. upper layer pyramidal neurons and later-born glial cells, mostly astrocytes. Pyramidal neurons traced in Fig. 5 were labeled by electroporating *PBCAG-Cytbow* (1 µg µl⁻¹) and *CAG-seCre* (0.1 µg µl⁻¹) without transposase in E15 Swiss embryos and brains were imaged at P21.

**Multiplex tract tracing with AAVs**. Three topographically distinct areas within the antero-lateral sensorimotor cortex were injected with AAV2/9 encoding FPs respectively emitting in the cyan (mTurquoise2[55] or mCerulean[60], yellow (mEYFP[56]) or red (tdTomato[57] or DsRed-express[61]) spectral range. Recombinant AAVs were based on a single-stranded (ss) AAV plasmid backbone with a CMV early enhancer/chicken β actin (CBA) promoter driving FP expression, except for DsRed-Express, driven by a CMV promoter. Detailed maps and sequences of the vectors are available upon request. Apart from the mCerulean vector, obtained from the Penn Vector Core (AV-9-PV1970), and the DsRed-Express vector generated as described in Kuhlman et al.[62], all recombinant AAVs were produced by co-transfecting the corresponding vector plasmids along with packaging and helper plasmids in HEK293T cells. The AAV particles were harvested by centrifugation and dialyzed. AAV vector titers were then determined using real-time qPCR and adjusted to 0.5–5×10¹³ vg ml⁻¹ by dilution in phosphate buffer saline (PBS) prior to injection. C57Bl6 female mice aged 11 weeks were anesthetized by inhalation of 2% isoflurane delivered with a constant oxygen flow (0.4 l min⁻¹). Dexamethasone (0.5 mg kg⁻¹) and ketoprofen (5 mg kg⁻¹) were administered subcutaneously for pre-emptive analgesia prior to the surgery and lidocaine (2-4 mg kg⁻¹) was applied preemptively intra-incisionally. Mice were mounted in a stereotaxic headframe (Kopf Instruments, 940 series). Stereotactic coordinates were identified for three topographically distinct brain areas within the antero-lateral sensorimotor cortex that were respectively injected with the three AAVs. An incision was made over the scalp, three small burr holes drilled in the skull and brain surface exposed at each site. Injections were performed using the following AAVs and coordinates along the medio-lateral (ML), antero-posterior (AP), and dorso-ventral (DV) axis (defined relative to Bregma for ML and AP coordinates, and with reference to the surface of the brain for DV): in the whole brain projection mapping experiment (Fig. 6), we injected AAV2/9-CBA-tdTomato-WPRE (ML 2.4 mm, right; AP + 2.7 mm; DV 1 mm, α = −30 °), AAV2/9-CBA-mTurq2-WPRE (ML 2.4 mm, right; AP + 2.2 mm; DV 1 mm, α = 0°) and AAV2/9-CBA-EYFP-WPRE (ML 2.4 mm, right; AP + 1.9 mm; DV 1 mm, α = −30 °). For the WM and THG imaging experiment (Fig. 7), we used AAV2/9-CMV-BI-DsRed-Express-WPRE (ML 2.4 mm, right; AP + 2.7 mm; DV 1 mm, α = −30°); AAV2/9-CAG(CB7)-CI-mCerulean-WPRE-rBG (ML 2.5 mm, right; AP + 2.3 mm; DV 0.7 mm, α = 0°) and AAV2/9-CBA-EYFP-WPRE (ML 2.7 mm, right; AP + 2.0 mm; DV 1 mm, α = −30°). A pulled glass pipette tip of 20–30 µm containing the AAV suspension was lowered into the brain and was delivered at a rate of 30 nl min⁻¹ using a Picospritzer (General Valve Corp) at each of the three sites; following each virus injection, the pipette was left in its location for 10 min to prevent backflow. After the injection, the incision was closed with 5/0 nylon suture thread (Ethilon Nylon Suture, Ethicon Inc. Germany), and animals were kept warm on a heating pad until complete recovery.

**ChroMS setup and image acquisition**. Imaging was performed on a lab-built laser scanning microscope. The excitation source used was a Ti:Sapphire+OPO chain (Chameleon Ultra2 and MPX, Coherent, CA, USA). The two synchronous output beams were combined using a dichroic filter (DMSP-1000, Thorlabs) and temporally synchronized using a long-range motorized delay line (Thorlabs) placed on the OPO beam path, as described in ref. [30]. The power of the TiS and OPO beams were controlled using motorized waveplates and polarizers. In multicolor experiments, efficiency of the 2c2P virtual green excitation was independently controlled by adjusting the motorized delay line[30]. Scanning was performed using galvanometric scanners (series 6215, Cambridge Technology, MA, USA). Excitation beams were focused into the sample using a water immersion objective (25×, 1.05NA, XLPLN25XWMP2, Olympus, Japan). Beam size at the rear pupil of the excitation objective and beam divergence were adjusted using two-lens telescopes, one on each beam path. To prevent photobleaching, an electronically controlled beam shutter was installed after beam recombination together with an electro-optical modulator blocking TiS excitation during laser fly-back in unidirectional scan mode.

A vibrating blade microtome (Tissue Vision, USA) was integrated into the upright microscope setup, together with long-range high-precision XYZ translation stage (Physik Instrumente, Germany/ Tissue Vision, USA) for mosaic imaging and serial z-sectioning. Scanning and acquisition were synchronized using lab-written LabVIEW software and a multichannel I/O board (PCI-6115, National Instruments, USA). Mosaic acquisition and tissue slicing were controlled by a custom version of the Orchestrator software (Tissue Vision, USA) synchronized with the acquisition software through a lab-written LabVIEW module. Signals were detected in the backward (epi) direction by photomultiplier tubes (SensTech, UK) and/or GaAsP detectors (H7422P-40, Hamamatsu, Japan) using lab-designed electronics enabling either counting or analog detection. Fluorescence and coherent signals were collected using a dichroic mirror (Semrock 751 nm, USA) and directed towards four independent detectors using dichroic mirrors (Chroma 405 nm, Semrock 521 and 561 nm). Bandpass filters were used in front of the detectors to collect blue (Semrock 475/25), green (Semrock 547/22), and red (Semrock 607/70) light. The latter red bandpass filter was removed for CARS imaging. Third harmonic generation was collected with a dichroic mirror (Chroma, 405 nm).

Sample mount consisted of a rotating slide holder fixed into an ~2 l plastic tank installed on the translating stage. The objective and microtome arm were immersed in the PBS-filled tank during the entire duration of acquisitions (see Supplementary Fig. 13).

For vibrational imaging, excitation wavelengths were set to 832/1090 nm. Otherwise, excitation wavelengths were set to 850/1100 nm. To minimize lateral chromatic aberration and field curvature, the scan system was designed using a Plössl-lens configuration[63], and optics coating were also chosen to minimize aberrations (see Supplementary Fig. 14). Individual tile size was 300–360 µm × 300-360 µm in multicolor experiments (Supplementary Fig. 14) and 200 µm × 200 µm in THG-CARS experiments. A 10% lateral overlap was used for mosaic reconstruction. In the tomography acquisition mode, imaging was typically performed 40 µm below the surface. In the continuous acquisition mode, 80–130-µm-thick stacks were acquired after each 60–110-µm-thick tissue section to allow 3D stitching. For robust tissue slicing, the frequency of the microtome vibrating blade was set to 60 Hz and slicing speed to 0.3-0.5 mm s⁻¹. Axial sampling were set to 1.8 µm (TiS) and 2.6 µm (OPO) with <0.5 µm of axial mismatch. Metrology and spatial beam coalignment were done using KTP nanoparticles[33]. Lateral sampling in continuous mode acquisitions was set to 0.4 µm pixel⁻¹ and 0.54 µm pixel⁻¹ for

the ChroMS/astrocytes (Figs. 3 and 4) and ChroMS/neuron tracing (Fig. 5) experiments, respectively. In tomography mode acquisitions, it was set to 0.55 μm pixel$^{-1}$, 0.8 μm pixel$^{-1}$, and 1.6 μm pixel$^{-1}$ for the ChroMS/Brainbow (Fig. 1), ChroMS/Triple AAV (Fig. 6), and ChroMS/Endogenous signals (Fig. 7) experiments, respectively. Multicolor pixel dwell time was typically 5 μs.

**Images pre-processing and stitching**. Image processing was done using Fiji[64] (NIH,USA) and Matlab (Mathworks, USA), and 3D renderings were generated using IMARIS (Bitplane, Switzerland). Individual tiles were first batch cropped to remove scanning edge artifacts. Flat-field correction was then applied to individual tiles prior to stitching to correct for illumination inhomogeneity in the field of view. Illumination profiles were computed for each channel by averaging all the tiles of the channel and applying a large kernel Gaussian blurring. The tiles were then divided by the normalized (maximum intensity) illumination profiles and merged into RGB composites. Mosaic stitching was done using the Grid/Collection Stitching Fiji plugin[65]. Tile positions were computed using cross-correlation between tiles. For ChroMS continuous reconstructions, we used a custom version of the same plugin allowing 3D cross-correlation calculations between blocks. Of note, the high image quality provided over the entire acquisition depth by two-photon imaging permitted efficient 3D stitching without the need of a supplementary registration channel.

**Linear unmixing**. In order to remove spectral bleedthrough, a linear unmixing procedure was applied as follows (see Supplementary Fig. 2). The colorimetry of each dataset was first characterized with a pool of representative > 500 regions of interest (ROIs) selected in color-labeled zones. The representative pool was plotted in a ternary color diagram and a linear transform was applied to stretch RGB values over the entire color diagram. Channel intensities were modeled as follow:

$$\begin{cases} R_{raw} = R_{corr} + a_{GR}.G_{corr} + a_{BR}.B_{corr} \\ G_{raw} = a_{RG}.R_{corr} + G_{corr} + a_{BG}.G_{corr} \\ B_{raw} = a_{RB}.R_{corr} + a_{GB}.G_{corr} + B_{corr} \end{cases}$$

In images of Brainbow mice (Fig. 1) or MAGIC markers (Figs. 2, 3, and 5), the $a_{XX}$ coefficients were determined using maximum red, green and blue intensity ROIs $R_{max}(r_r,g_r,b_r)$, $G_{max}(r_g,g_g,b_g)$, $B_{max}(r_b,g_b,b_b)$ and by projecting the maximum blue intensity ROI to the 'pure blue' point in the ternary diagram. The stretching coefficients were then obtained by inverting the system associated matrix. The same procedure was used for unmixing the AAV-labeled datasets (Fig. 6), using barycentres of the ROI distributions obtained for each AAV (red, green, and blue) instead of maxima.

**Astroglial network reconstruction and segmentation**. Analyses of cortical protoplasmic astroglial cell distribution and contacts (Fig. 4) were done using Fiji[64] and Matlab (Mathworks, USA). A partial rendering of the astroglial network was obtained by reconstructing 3D color-clusters of astrocytes labeled with MAGIC markers. A 3D color-cluster was defined as an ensemble of astrocytes expressing a same color combination and continuously in contact with each other (see Fig. 4a). Individual astrocytes were considered as color-clusters of size 1. Within each cluster, the positions of astrocyte somata were pointed, together with the positions of negatively contrasted contacting cell bodies of 4–6 μm diameter size. Cell bodies were discriminated from transverse blood vessel sections using adjacent upper and lower slices within the z-stack. Both cell bodies engulfed within protoplasmic astrocytic domains and those located at the astrocytes' periphery were included in this analysis. The ratio of contacted cells over labeled astrocytes was computed for all clusters, except pial. For the analysis of astrocyte contacted cell number in different cortical layers (Fig. 4c), only clusters of size ≤3 which were entirely comprised within an identified cortical layer were considered. Cortical layers were identified based on morphological landmarks (see Supplementary Fig. 4b): layer 2/3 was identified by the neuronal labeling resulting from E15 electroporation[66]; layer 5a was discriminated using visible projections from layer 2/3 pyramidal neurons[66]; and layers 5b and 6 were discriminated based on cell density, size and distance to the corpus callosum. The pial surface was automatically segmented using intensity thresholding and morphometric boundaries detection. The segmented surface was than downscaled and fitted with linear interpolation for 3D visualization. 3D segmentation of individual astrocyte domains was done on isolated or color-segregated cells using IMARIS 8.4v (Bitplane, Switzerland). For each astrocyte, semi-automatic iso-intensity contours were drawn in all slices of the corresponding z-stack (see Supplementary Fig. 4a). When the iso-intensity contour tool did not provide accurate segmentation (low color SNR), manually defined contours were drawn. Only astrocytes that could be unambiguously attributed to a specific cortical layer (as defined above) were considered for layer analyses. To analyze the relation between contacted cells number and astrocyte domain volume (Fig. 4e), only individual astrocytes and pairs comprised within an identified cortical layer of which contacting cells had been pointed were considered (the domain volume of astrocyte belonging to a pair being approximated as half of the total volume of the pair).

**Statistical analysis**. Statistical analysis of astrocyte volume distributions in different cortical layers was performed using GraphPad Prism software. Sample size being unequal across all the categories and without any assumption on the distributions, a non-parametric one-way anova test was first performed (Kruskal-Wallis), followed by a post hoc multiple comparisons test (Dunn).

**Astrocytes tridimensional interface analysis**. For each astrocyte pair analyzed, the interface between the two-color segregated astrocytes ($A_1$, $A_2$) was segmented using Fiji[64] resulting in a discrete set of 3D coordinates {$x_I,y_I,z_I$} in the XYZ acquisition frame. The 3D positions {$x_{A1},y_{A1},z_{A1}$} and {$x_{A2},y_{A2},z_{A2}$} of $A_1$ and $A_2$ somatas defined the Voronoi axis, $\overrightarrow{A_1 A_2}$ being set as axis orientation vector. The Voronoi plane was defined as the plane perpendicular to the Voronoi axis and passing through the astrocyte pair midpoint $M$ defined as the center of the [$A_1 A_2$] segment. Interface 3D coordinates were then expressed in the Voronoi frame and fitted with a 2D polynomial function (see Supplementary Fig. S5). The resulting fitting plane was defined as the interface plane. The angle between the Voronoi plane and the interface plane was computed (orientation parameter) together with the ratio of interface points above the Voronoi plane (bias parameter). The interface plane and a corresponding normal vector were used to define the interface frame. 3D interface coordinates were in turn expressed in the interface frame and fitted with a biharmonic surface model. Interfaces were then semi-automatically detoured and amplitude and gradient maps were generated. All interface computations were done using Matlab (Mathworks, USA).

**Neuron tracing**. Neurite tracing in Fig. 5 was done using the Imaris Filament Tracer tool, IMARIS v9.2 (Bitplane, Switzerland).

**Quantitative topography with triple-AAV projection labeling**. A pipeline for quantifying neuronal projection color maps was adapted from previously described methods designed for serial two-photon microscopy-based monochrome image analysis[23,24,27], as follows. After spectral unmixing, fluorescence signal was separated from background for each channel. Square root transform followed by histogram matching was applied to intensity images for contrast enhancement and standardization prior to segmentation. Florescence signal was then separated from background using an intensity-based clustering algorithm (multi-Otsu)[67] combined with connected component labeling (see Supplementary Fig. 8). For each channel, the segmented binary mask was transformed into a color-exclusive mask by excluding signal pixels shared with other channels. Color-exclusive projection strength maps were then generated by computing the ratio of signal pixels over background in 30 × 30 pixel-size non-overlapping adjacent super-pixels. Finally, red, green and blue projection strength maps were combined onto RGB interdigitation maps (see Supplementary Fig. 9). Projection strength maps were generated together with confidence maps to account for quantification errors due to multi-channel pixel exclusion. Confidence maps were computed as the ratio of exclusive signal pixels over the total number of signal pixels in the 30 × 30 pixel-size super-pixels (see Supplementary Fig. 10). For quantitative analyses of cortical projections in the striatum (Fig. 6i, j), the striatal area was first manually detoured based on the Allen Brain reference atlas and associated interdigitation maps were generated (see Supplementary Movie 14). Within each super-pixel, the relative contribution of red, green, and blue projection strength was calculated and the distance to seven reference points in the RGB color space ([1 0 0], [0 1 0], [0 0 1], [0.5 0.5 0], [0.5 0 0.5], [0.5 0 0.5], [0.33 0.33 0.33]) was calculated. Each super-pixel was classified in the category minimizing the distance in the RGB color space. This enabled to identify super-pixels containing single, dual and triple projections. For the analysis of cortical axons arbors in the caudoputamen (Fig. 6i), masks encompassing arborizing axons and terminals and excluding fiber tracts and ventral striatal areas were manually detoured. For the analysis of corticofugal tracts (Fig. 6j), the complementary masks encompassing passing fibers were used. Since tracts exhibited systematically high signal levels at their center, this resulted in residual spectral bleed-through despite the linear unmixing procedure which was manually removed. Non-exclusive interdigitation masks were considered for this analysis of fiber tract interdigitation, since it probes larger structures than that of axonal arbors interdigitation. All the analyses were implemented using custom-written Matlab (Mathworks,USA) scripts and Fiji[64] (NIH,USA).

## Data availability

The brain-wide Brainbow dataset (Fig. 1) and the analyzed 3D cortical dataset (Figs. 3 and 4) are publicly available as resource datasets in the Image Data Resource (IDR)[68] [https://idr.openmicroscopy.org] under accession number idr0048. Raw unprocessed data is available upon request.

## Code availability

Custom code used for data analysis and image processing pipeline is available upon request

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

## Acknowledgements

We thank Minh-Son Phan, Samuel Tozer, and Z. Josh Huang for scientific discussions and comments on the manuscript. We thank TissueVision for providing technical advice for the implementation and customization of the block-face system. We thank Bastien Binet from Polytechnique Polymedia center for his generous help in the creation of 3D graphics. We thank IdV core imaging, animal experimentation and histology facilities for technical assistance. This work was supported by fellowships from Université Paris-Saclay (Initiatives Doctorales Interdisciplinaires) to L.A., Ecole des Neurosciences de Paris to K.M., Région Ile-de-France and Fondation ARC pour la Recherche sur le Cancer to S.C. by Agence Nationale de la Recherche under contracts ANR-11-EQPX-0029 (Equipex Morphoscope2), ANR-10-INBS-04 (France BioImaging), and ANR-10-LABX-65 (LabEx LifeSenses), by Fondation pour la Recherche Médicale (grant DBI20141231328), by Target ALS and by the European Research Council (ERC-CoG 649117).

## Author contributions

E.B., J.L., L.A., K.M., W.S., S.T., and J.Lich designed the study. K.M. and S.T. performed initial tests on a prototype tissue slicer. L.A., E.B., P.M., J.M.S., X.S., and W.S. implemented the ChroMS imaging setup. J.L., K.L., K.M., S.C., and A.B. designed and implemented labeling strategies. K.L., S.C., and K.M., respectively, prepared Cytbow-, MAGIC markers-, and AAV-labeled samples. LA prepared samples for block-face imaging and performed all experiments, with assistance from S.C. for astrocytes experiments. I.A.C. and A.C. implemented stitching and data management strategies. L.A. processed the large volume datasets and analyzed the data. L.A., J.L., and E.B. wrote the manuscript with the assistance of all other authors.

## Additional information

**Competing interests:** E.B., P.M., and W.S. are co-authors of a patent that describes multicolor two-photon excitation by wavelength mixing (FR1250990). The remaining authors declare no competing interests.

