## [Peer Review File · Nature Communications]

Reviewers' comments:

Reviewer #1 (Remarks to the Author):

Abdeladim and colleagues describe an application of serial two-photon tomography using the commercial instrument TissueCyte with the additional twist that they apply two femtosecond lasers at 2 wavelengths, e.g. 850 and 1100 nm. While the experiments are nicely done, the technical advance of using 2 lasers with the TissueCyte instrument is incremental and the scientific information in the paper is minimal.

The authors first show images from the Brainbow mouse brains, but their quantification of these data is rudimentary and limited to general statements, such as visualization of "fine neural processes, such as Purkinje cell dendrites and axons in the cerebellum". Can the authors use their imaging to trace and reconstruct entire neuronal morphologies in the brainbow labeling?

In the second application, the authors image 3D distribution of astrocytes and report "significant volume variation between layer 4 astrocytes and astrocytes located in projection layer 5a", "imbalance in the positioning of the limits of astrocyte cytoplasmic domains with respect to the median plane" and "the tiling of the cerebral cortex by astroglial cells deviates from an equiparted Voronoi geometry." However, these results / observations are left at the anecdotal level of small volumes analyzed and the authors state that "complete anatomical characterization of the astroglial network is beyond the scope of the present study", though it is not clear why this should be so. A demonstration of a complete whole brain analysis and public depository of such a dataset would greatly add to the manuscript's incremental methodological advance.

Finally, the authors show 3-color imaging of AAV-labeled neurons in the mouse brain, but again without demonstrating any scientific results beyond generating pretty images. Taken together, the paper in the current state may be more appropriate for a specialized optics journal. Alternatively, the authors could extend some of their preliminary experiments to reveal at least some novel scientific information and/or generate datasets that could be used as a resource by the neuroscience community.

Reviewer #2 (Remarks to the Author):

This paper reported a multimodality multiphoton microscope by integrating automated serial tissue sectioning and several nonlinear imaging modalities including two-photon, THG and CARS. The whole brain images are beautiful and show the structure details. Also, this paper addressed the chromatic aberration by using SHG nanocrystals to ensure the two beams spatially overlapped. However, the technology novelty is mitigated since similar technologies have been reported [1,2]. The method for quantifying chromatic aberration was previously demonstrated as well [3]. Therefore, biological discoveries enabled by the Chrom-SMP are anticipated and need to be emphasized in the paper.

[1] P. Mahou, M. Zimmerley, K. Loulier, K. S. Matho, G. Labroille, X. Morin, W. Supatto, J. Livet, D. Débarre, E. Beaurepaire, "Multicolor two-photon tissue imaging by wavelength mixing" *Nat. Methods* 2012, 9, 815.

[2] T. Ragan, L. R. Kadiri, K. U. Venkataraju, K. Bahlmann, J. Sutin, J. Taranda, I. Arganda-Carreras, Y. Kim, H. S. Seung, and P. Osten, "Serial two-photon tomography for automated ex vivo mouse brain imaging," *Nat. Methods* 9(3), 255–258 (2012).

[3] Mahou, P., Malkinson, G., Chaudan, É., Gacoin, T., Beaurepaire, E., & Supatto, W. Metrology of Multiphoton Microscopes Using Second Harmonic Generation Nanoprobes. *Small*, 13(42), 1–11 (2017).

2. In page 2, the statement of obtaining label-free signals simultaneously is ambiguous because the system used 832/1090 nm to excite CARS signals and 840/1100 nm to excite THG and two-photon signals. Switching excitation wavelengths is required and takes time while acquiring different modalities' signals.
3. In page 5, the methods for correcting the lateral and axial chromatic aberration need to be Illustrated or cite the related references.
4. In page 13, more discussions on the advantages of integrating CARS and THG are encouraged as the paper utilized both modalities targeting on lipid. Furthermore, one advantage of CARS imaging is its rich chemical specificity while probing the various Raman vibrational modes. How CARS being integrated to Chrom-SMP can enable new discovery in whole brain imaging also needs to be discussed.
5. The limitation of the image acquisition time for whole brain mapping is obvious. Serial tissue sectioning sacrifices sample and requires extra time for sectioning the layers by using blade microtome. On page 16 paragraph 3, it would be beneficial to point out how much room of the imaging speed may be potentially improved.

Reviewer #3 (Remarks to the Author):

There has been a number of papers describing whole brain imaging by serial sectioning and 3D reconstruction. Expanding the color palette will certainly make serial whole brain imaging more powerful, especially for mapping the connectome of Brainbow and other multi-color mice. Here the authors accomplish this by implementing an elegant wavelength mixing technique that combines two femtosecond pulse trains spatially and temporally to generate a third, "virtual" color. Although the wavelength mixing technique has been published in an earlier paper by the same authors (Nature Methods 2012), the work is still state of the art; the manuscript is well written and beautifully illustrated. In my opinion, it is suitable for publication in Nature Communications. However, there are a few concerns:

1. What is the scale bar in Fig. 1a (lateral PSF)? What is the consequence in terms of color fidelity when there is a residual lateral chromatic shift of 0.6 μm as shown in Fig. 1b? I would expect that as the beam moves off axis, the "virtual" excitation would become less efficient due to the reduced overlap of the two PSFs. I would also expect that the 0.6 μm separation is not negligible compared to the size of the PSFs. It will be useful if the authors could supply an additional graph, similar to the lower right hand panel of Fig 1b but showing the efficiency of two-photon excitation at the "virtual" transition wavelength.
2. Related to #1 above, according to the Methods description (p.19), flat field correction is applied to each of the three channels (RGB) independently. Again please discuss the consequence of such correction in terms of color fidelity when the three channels are merged.
3. Also in Fig 1b, why is the axial chromatic shift not zero at the center of the field? Since each arm of the two laser beams has a telescope, it should be possible to achieve perfect overlap at the center of the field. Is there a reason why the is not done?
4. Throughout the manuscript, the terms "contact" and "interaction" seem to be used interchangeably. I suggest the authors avoid the use of "interaction", since without live or functional imaging data, it is impossible to know if the cells in physical contact are really interacting.
5. In my understanding, psec laser sources are more optimal for CARS imaging, as the bandwidth of the laser matches the linewidth of the vibrational transition. Could the authors comment on the efficiency of fsec CARS in this setup? Also please comment on the modification that would be

needed to allow imaging other molecular transitions besides the C-H stretch commonly used for imaging lipids (which as shown by the authors can also be imaged by THG).

Minor concerns:

6. Please clarify if Fig 1, panel a2 is a maximum intensity projection. Also are b1-b4 also MIPs? It will be useful if the authors provide the thickness of the MIPs.

7. Bottom of p. 13, "THG-CARS imaging obtained using mixed 832/1040 nm excitation" is at odds with the Fig 6 caption (b) ... excitation wavelength set to 832/1090 nm. I believe 1090 is the correct number and 1040 is an error. Please correct the typo.

8. Top of p. 19, "The objective and vibratome arm were immersed in the PBS-filled tank during the entire duration of acquisitions. Please provide more details on how this is implemented (perhaps including a photo of the actual setup).

Point-to-point response to the reviewers' comments

Manuscript NCOMMS-18-06441 – Multicolor multiscale brain imaging with chromatic multiphoton serial microscopy

Summary of the main changes:

- Text changes are indicated in purple;
- Designed a new experiment (Fig 5, movies M9, M10) demonstrating color-based dense neuronal tracing in a thick cortical sample;
- Doubled astrocyte morphometric analyses and drew new conclusions on cortical astrocyte heterogeneity (Fig 4);
- Additional analyses of interdigitation in the striatum (Figs 6, S11, S12, movie M14);
- Additional information on the setup, which is not a commercial instrument (fig S13, S14);
- Public online availability of two datasets and related analyses.
- New flythrough movies illustrating the unmatched quality of our data (M7, M9, M10).

Reviewers' comments:

Reviewer #1 (Remarks to the Author): Abdeladim and colleagues describe an application of serial two-photon tomography using the commercial instrument TissueCyte with the additional twist that they apply two femtosecond lasers at 2 wavelengths, e.g. 850 and 1100 nm. While the experiments are nicely done, the technical advance of using 2 lasers with the TissueCyte instrument is incremental and the scientific information in the paper is minimal. The authors first show images from the Brainbow mouse brains, but their quantification of these data is rudimentary and limited to general statements, such as visualization of “fine neural processes, such as Purkinje cell dendrites and axons in the cerebellum”. Can the authors use their imaging to trace and reconstruct entire neuronal morphologies in the brainbow labeling?

In the second application, the authors image 3D distribution of astrocytes and report “significant volume variation between layer 4 astrocytes and astrocytes located in projection layer 5a”, “imbalance in the positioning of the limits of astrocyte cytoplasmic domains with respect to the median plane” and “the tiling of the cerebral cortex by astroglial cells deviates from an equiparted Voronoi geometry.” However, these results / observations are left at the anecdotal level of small volumes analyzed and the authors state that “complete anatomical characterization of the astroglial network is beyond the scope of the present study “, though it is not clear why this should be so. A demonstration of a complete whole brain analysis and public depository of such a dataset would greatly add to the manuscript's incremental methodological advance.

Finally, the authors show 3-color imaging of AAV-labeled neurons in the mouse brain, but again without demonstrating any scientific results beyond generating pretty images. Taken together, the paper in the current state may be more appropriate for a specialized optics journal. Alternatively, the authors could extend some of their preliminary experiments to reveal at least some novel scientific information and/or generate datasets that could be used as a resource by the neuroscience community.

We thank the reviewer for his/her compliments on our experimental work. We have included substantial additions in our revised manuscript to answer the requests made above, but first we would like to address the reviewer's comments on the technical advance that our work represents.

We realized that one point was not clear enough and we improved our manuscript to clarify it: the setup used in this study is not a “commercial TissueCyte instrument with 2 lasers”. **Our imaging system is a custom-built microscope** (frame, optics, beam scanning system, detection electronics) to which we have integrated TissueCyte elements bought from the TissueVision company (the vibratome arm, large-scale XYZ stages, and a sample holding cuvette where we redesigned the sample mounting system for more flexibility). We used our own Labview software for acquisition of XYZ stacks, and interfaced it with a stripped-down version of TissueVision’s software for acquisition of mosaics. In the revision version, we have integrated a new figure S13 showing pictures of the mounting stage and information about the beam scanning system; the custom-built microscope frame is visible behind the sample. We had to build a setup because there is currently no commercial alternative.

We also want to stress that **our platform is the very first to enable the generation of chromatically corrected large volume multicolor datasets**, despite major efforts in large-volume microscopy and the plethora of clearing protocols published in recent years. In other terms, since Brainbow and other combinatorial labeling strategies require a stringent control of chromatic aberration with depth, no multicolor dataset of more than a few hundreds of microns has been published in the literature so far. Indeed, recent articles on combinatorial labeling strategies (Sakaguchi et al, eLife 2018) state explicitly that chromatic aberration remains a major bottleneck for thick tissue imaging. Besides, our paper also demonstrates 3-color neural projection imaging, which is of obvious interest and has never been achieved before despite the advances stated above. Together, the unique capabilities cited above demonstrate the technical advances of the setup are not incremental.

In our revised version, we incorporated the reviewer’s suggestions to further improve the paper and extract more biological insight from our data. Notably:

(A) We doubled the number of segmented astrocytes in our analysis of astrocyte domain territories and drew novel conclusions on layer-dependent cortical astrocyte heterogeneity (see Figure 4d-e). We demonstrate **significant heterogeneity of the territorial volume of layer 4-6 astrocytes**. Furthermore, our analysis demonstrates that astrocyte volume and number of engulfed cells are not systematically correlated amongst individual astrocytes, but that **each cortical layer exhibits characteristic values for these two parameters**. We point out that **3D** segmentation of more than a hundred of astrocytes within the same brain is a so far unmatched and critical performance (recent articles in the field are based on smaller total number of cells, and present distorted measures due to the tissue clearing protocol used). Moreover this number is not limited by the ChroMS technique and can be upscaled by refining labeling strategies (e.g. to mark a larger fraction of astrocytes). We would also like to highlight that evidence for heterogeneity within the cortical protoplasmic astrocyte population has started to emerge in the field only very recently (Lanjakornsiripan et al, Nat Commun 2018; Bayraktar et al, Biorxiv 2018). Therefore the novel measurements provided in our paper, together with the dataset and analyses which will be accessible online will constitute a valuable contribution to this emerging question.

(B) For this revised version of our manuscript, we have designed and added a new experiment demonstrating dense neuronal tracing in a thick portion of mouse brain cortex (see new Figure 5). **This experiment is to our knowledge the first demonstration of dense neuronal tracing across millimetric depths.**

Going beyond our work and achieving reconstruction of neuronal morphologies across the entire brain will require to (i) increase the speed of the ChroMS system (cf. discussion); ii) improve the labeling to achieve brighter labeling of thin axons; and (iii) as importantly, develop automated

neuron tracing algorithms. The outcome of these synergistic developments should enable the reconstruction of thousands of neurons within the same brain, and ChromS imaging technology is the first milestone to reach this outstanding goal.

(C) We have also further demonstrated how quantitative interdigitation analysis of multiple projections within the same brain enables to generate novel anatomical insights by analyzing projection topography in the striatum (Fig 6, S11, S12). This approach will be of immediate relevance when combined with recent developments on cell type- and/or layer-specific viral vector labels.

(D) Finally, the IDR platform (<https://idr.openmicroscopy.org/about/>) has accepted to host our datasets. We thank the reviewer for suggesting that we make such public deposition.

Reviewer #2 (Remarks to the Author): This paper reported a multimodality multiphoton microscope by integrating automated serial tissue sectioning and several nonlinear imaging modalities including two-photon, THG and CARS. The whole brain images are beautiful and show the structure details. Also, this paper addressed the chromatic aberration by using SHG nanocrystals to ensure the two beams spatially overlapped. However, the technology novelty is mitigated since similar technologies have been reported [1,2]. The method for quantifying chromatic aberration was previously demonstrated as well [3]. Therefore, biological discoveries enabled by the Chrom-SMP are anticipated and need to be emphasized in the paper.

[1] P. Mahou, M. Zimmerley, K. Loulier, K. S. Matho, G. Labroille, X. Morin, W. Supatto, J. Livet, D. Débarre, E. Beaufrepaire, "Multicolor two-photon tissue imaging by wavelength mixing" *Nat. Methods* 2012, 9, 815.

[2] T. Ragan, L. R. Kadiri, K. U. Venkataraju, K. Bahlmann, J. Sutin, J. Taranda, I. Arganda-Carreras, Y. Kim, H. S. Seung, and P. Osten, "Serial two-photon tomography for automated ex vivo mouse brain imaging," *Nat. Methods* 9(3), 255–258 (2012).

[3] Mahou, P., Malkinson, G., Chaudan, É., Gacoin, T., Beaufrepaire, E., & Supatto, W. Metrology of Multiphoton Microscopes Using Second Harmonic Generation Nanoprobes. *Small*, 13(42), 1–11 (2017).

We thank the reviewer for his/her compliments on our images. The Reviewer's comments rightfully underlines that our present work leverages technological development previously made by our groups and others. We however combine these techniques in a synergistic and unprecedented way to address an important bottleneck in multicolor imaging. This approach enabled us to generate **the very first large volume, micron-scale resolution multicolor image datasets** with quality suitable for color-based analyses (whatever the technique used). Following the reviewer's suggestion, in this revised version of the manuscript, we have specifically focused our efforts on emphasizing the biological relevance of our technique –see on this aspect points (A-C) in the reply to reviewer #1 above and Figure 4c-e, 6i-j, S11 and S12.

We also addressed the additional technical issues raised by Reviewer #2 and we thank him/her for the helpful comments, which helped us clarifying our statements in the text (see below).

2. In page 2, the statement of obtaining label-free signals simultaneously is ambiguous because the system used 832/1090 nm to excite CARS signals and 840/1100 nm to excite THG and two-photon signals. Switching excitation wavelengths is required and takes time while acquiring different modalities' signals.

Indeed, our acquisition scheme does not include excitation wavelength switching. In the set of presented experiments we either used the 850/1100 nm or the 832/1090 nm excitation configuration. The first configuration enables Brainbow + THG imaging, and the second configuration enables CARS+THG (+SHG/or green fluorescence) imaging. We have clarified this statement in the text and in Figure S13 which summarizes the contrast modalities accessible to our methodology.

3. In page 5, the methods for correcting the lateral and axial chromatic aberration need to be illustrated or cite the related references.

We have added an additional supplementary figure with the Zemax simulations used to design our scanning system in order to minimize lateral chromatic aberration (together with the related reference, Negrean et al cited in the methods section). Figure 2 shows then how we estimate chromatic aberration across a single field of view, ensuring that it does not exceed 0.6 μm across the entire imaged volume. We added a reference to Mahou, Small 2017, where the methodology is described. As suggested by reviewer #3, we present in Figure S14 a calculation of the wavelength mixing efficiency across the field of view (estimated from the measured chromatism), and the measured overall field flatness in the green channel (resulting from excitation and detection heterogeneity).

4. In page 13, more discussions on the advantages of integrating CARS and THG are encouraged as the paper utilized both modalities targeting on lipid. Furthermore, one advantage of CARS imaging is its rich chemical specificity while probing the various Raman vibrational modes. How CARS being integrated to Chrom-SMP can enable new discovery in whole brain imaging also needs to be discussed.

We agree with Reviewer #2 that the novel ability provided by ChromS imaging to record CARS/THG signals over entire brains will deserve an extensive study in itself. However, the focus of the current paper is more on the possibilities offered by large-scale brainbow imaging, which is of more immediate interest for connectomics, morphological, and clonal analyses.

We clarified our statements on CARS and THG, but prefer to leave these experiments as a 'simple' perspective opened by our current study. We would like however to make here a few comments in reply to the reviewer's questions:

CARS and THG can both be used to highlight myelin in brain tissue. However these two modalities rely on different contrast mechanisms and phase-matching conditions. THG is generally viewed as less specific. It highlights interfaces and is sensitive to the polarizability of CC bonds in lipid tails, whereas CARS at 2845 cm^{-1} probes CH bonds. Consequently, the two images provide complementary information on the microstructure of multilamellar lipidic objects, as e.g. illustrated in Figure 5a of Zimmerley et al, PRX 2013. We mention this point in our revised version. One perspective of combining large-scale THG-CARS imaging may therefore be to reveal unsuspected structural information in myelinated fibers. As a follow-up to the work presented here, it will indeed be very informative to perform a systematic comparison of CARS/THG images recorded in different brain locations, possibly with different excitation NAs. Finally, we agree with the reviewer that CARS can provide richer, chemically-specific information through the probing of several vibrational modes. However our system in its current implementation can probe only one mode at a time; therefore it can essentially be used to map lipidic regions. Here again, the ChromS technology opens the path to additional studies which are beyond the scope of the current work.

5. The limitation of the image acquisition time for whole brain mapping is obvious. Serial tissue sectioning sacrifices sample and requires extra time for sectioning the layers by using blade microtome. On page 16 paragraph 3, it would be beneficial to point out how much room of the imaging speed may be potentially improved.

We agree. Recent work on monochrome STP imaging (Economo et al, eLife 2016) has shown that, by combining intense AAV labelling, fast scanning and contour detection, an entire brain can be recorded in 7-10 days with continuous micrometer-scale sampling. The same can be applied to the ChroMS scheme. We have included a more explicit statement on this aspect in the discussion. However, as also stated in our discussion, the most time-limiting step in large volume microscopy studies is currently the image analysis and data mining part rather than the acquisition step.

Reviewer #3 (Remarks to the Author): There has been a number of papers describing whole brain imaging by serial sectioning and 3D reconstruction. Expanding the color palette will certainly make serial whole brain imaging more powerful, especially for mapping the connectome of Brainbow and other multi-color mice. Here the authors accomplish this by implementing an elegant wavelength mixing technique that combines two femtosecond pulse trains spatially and temporally to generate a third, "virtual" color. Although the wavelength mixing technique has been published in an earlier paper by the same authors (Nature Methods 2012), the work is still state of the art; the manuscript is well written and beautifully illustrated. In my opinion, it is suitable for publication in Nature Communications. However, there are a few concerns:

We thank the reviewer for his/her appreciation of the relevance of our work in the field. Below our comments on how we addressed the remaining concerns:

1. What is the scale bar in Fig. 1a (lateral PSF)? What is the consequence in terms of color fidelity when there is a residual lateral chromatic shift of 0.6 μm as shown in Fig. 1b? I would expect that as the beam moves off axis, the "virtual" excitation would become less efficient due to the reduced overlap of the two PSFs. I would also expect that the 0.6 μm separation is not negligible compared to the size of the PSFs. It will be useful if the authors could supply an additional graph, similar to the lower right hand panel of Fig 1b but showing the efficiency of two-photon excitation at the "virtual" transition wavelength.

We have added the scale bar value in the corresponding caption.

We have added in the revised version of Figure 2 a calculation of the mixed-wavelength excitation efficiency based on the measured chromatic aberration map. Indeed, this efficiency decreases with increasing spatial mismatch between the beams. In our experiments, we calibrate the overall variation in excitation and detection across the field of view for each channel, and use these data to perform a flat-field correction of the images. The procedure is described in the 'image analysis' paragraph of the Methods section.

2. Related to #1 above, according to the Methods description (p.19), flat field correction is applied to each of the three channels (RGB) independently. Again please discuss the consequence of such correction in terms of color fidelity when the three channels are merged.

Applying flat-field correction to each of the three channels is indeed necessary to ensure color fidelity and enabled us to perform color-based analyses across mosaic images. We have added in Supp Fig S2 (pre-processing steps) an image illustrating the effect of flat-field correction. Also, as mentioned above, we have added in Supp Fig S14 a calculation of the wavelength mixing efficiency across the field, based on the measured chromatic aberration.

3. Also in Fig 1b, why is the axial chromatic shift not zero at the center of the field? Since each arm of the two laser beams has a telescope, it should be possible to achieve perfect overlap at the center of the field. Is there a reason why the is not done?

What is required to ensure high-quality multicolor microscopy is to obtain simultaneously the two following conditions: (i) the two beams should effectively fill the objective entrance pupil to optimize resolution, and (ii) the two beams should axially overlap with a precision better than approximately half the axial resolution. We in fact use 2 telescopes in our system for that purpose. In practice, we ensured that these conditions were met for all the data presented in this article. Usually, the axial match was even better than what is illustrated in Fig 2a. The curves presented are just a representative case, where the slight mismatch is convenient for distinguishing the two profiles.

4. Throughout the manuscript, the terms "contact" and "interaction" seem to be used interchangeably. I suggest the authors avoid the use of "interaction", since without live or functional imaging data, it is impossible to know if the cells in physical contact are really interacting.

We corrected this. Our terminology referred to a commonly used way to describe neighboring astrocyte territories (Halassa et al, J Neurosci 2007; Lopez-Hidalgo et al, J Comp Neurol 2016), but the use of contact instead of interaction is certainly more appropriate for a broad readership.

5. In my understanding, psec laser sources are more optimal for CARS imaging, as the bandwidth of the laser matches the linewidth of the vibrational transition. Could the authors comment on the efficiency of fsec CARS in this setup? Also please comment on the modification that would be needed to allow imaging other molecular transitions besides the C-H stretch commonly used for imaging lipids (which as shown by the authors can also be imaged by THG).

It is correct that psec excitation is more optimal for CARS imaging, because it result in a better signal-to-background ratio. Using fsec excitation reduces the chemical sensitivity and specificity. However, it works well for mapping dense lipidic structures, and provides a convenient way to combine CARS imaging with over multiphoton modalities such as SHG and 2PEF, see e.g. Chen et al, Opt Exp 17, 1282 (2009).

Imaging another vibration mode on our setup can be realized by simply changing the excitation wavelengths in order to match the targeted mode. However, achieving spectral CARS imaging of non-CH modes with an optimal contrast would require to implement additional techniques, such as the spectral focusing approach (Hellerer et al, App Phys Lett 2004).

In its current implementation, our setup enables to acquire simultaneous (CH₂)CARS and THG images, which can provide complementary microstructural information on lipidic objects, as illustrated in Figure 5a of Zimmerley et al, PRX 2013. See also the additional comments above (answer to Reviewer #2, point #4).

Minor concerns:

6. Please clarify if Fig 1, panel a2 is a maximum intensity projection. Also are b1-b4 also MIPs? It will be useful if the authors provide the thickness of the MIPs.

Presumably the comment refers to Figure 3. We have clarified that in the figure caption.

7. Bottom of p. 13, "THG-CARS imaging obtained using mixed 832/1040 nm excitation" is at odds with the Fig 6 caption (b) ... excitation wavelength set to 832/1090 nm. I believe 1090 is the correct number and 1040 is an error. Please correct the typo.

We have corrected the typo.

8. Top of p. 19, "The objective and vibratome arm were immersed in the PBS-filled tank during the entire duration of acquisitions. Please provide more details on how this is implemented (perhaps including a photo of the actual setup).

As suggested by the reviewer, we provide in this revised version a new supplementary Figure S13 showing pictures of the setup.

REVIEWERS' COMMENTS:

Reviewer #1 (Remarks to the Author):

the authors have significantly improved the manuscript and answered my main concerns. I have no further comments.

Reviewer #2 (Remarks to the Author):

The authors has significantly improved their manuscript. I have no further comments.

Reviewer #3 (Remarks to the Author):

I appreciate the additional information provided by the authors in response to the previous review. Minimizing chromatic aberration for large scale multicolor tissue imaging with submicron precision and channel co-registration is a nontrivial task, and the authors have taken important step in meeting this challenge. The wavelength mixing technique is well-suited for this purpose, and in my opinion, the manuscript is of significant interest to the multiphoton microscopy field. However, I have one additional concern. In the newly added Figure S2 d-e, showing the effect of flat field correction, can the authors explain why there are pronounced color changes before and after correction at multiple locations in the image, not just limited to the periphery of the field? Please also provide details of flat field correction in the Methods section, as this is a critical step in ensuring color fidelity.

Point-to-point response to the reviewers' comments

Manuscript NCOMMS-18-06441 Revision 2 – Multicolor multiscale brain imaging with chromatic multiphoton serial microscopy

Reviewers' comments:

Reviewer #1 (Remarks to the Author): the authors have significantly improved the manuscript and answered my main concerns. I have no further comments.

We thank the reviewer for his/her appreciation.

Reviewer #2 (Remarks to the Author): The authors has significantly improved their manuscript. I have no further comments.

We thank the reviewer for his/her appreciation.

Reviewer #3 (Remarks to the Author): I appreciate the additional information provided by the authors in response to the previous review. Minimizing chromatic aberration for large scale multicolor tissue imaging with submicron precision and channel co-registration is a nontrivial task, and the authors have taken important step in meeting this challenge. The wavelength mixing technique is well-suited for this purpose, and in my opinion, the manuscript is of significant interest to the multiphoton microscopy field.

We thank the reviewer for his/her positive comments.

However, I have one additional concern. In the newly added Figure S2 d-e, showing the effect of flat field correction, can the authors explain why there are pronounced color changes before and after correction at multiple locations in the image, not just limited to the periphery of the field?

There are such changes at several locations simply because the image displayed in Supp Figure 2d-e is a mosaic of approximately 4x3 tiles, and not a single field-of-view. A grid pattern is detectable as darker regions in the background, corresponding to the field periphery of adjacent individual tiles. The fact that this images is a mosaic is mentioned as follows in the figure legend: "(d-e) Mosaics from Brainbow labeled cortical tissue before (d) and after (e) combined flat-field correction and color processing".

Please also provide details of flat field correction in the Methods section, as this is a critical step in ensuring color fidelity.

The flat-field correction procedure is described in the "Images pre-processing and stitching" Methods subsection, as follows: "Flat-field correction was then applied to individual tiles prior to stitching to correct for illumination inhomogeneity in the field of view. Illumination profiles were computed for each channel by averaging all the tiles of the channel and applying a large kernel Gaussian blurring. The tiles were then divided by the normalized (maximum intensity) illumination profiles and merged into RGB composites."